# One Risk to Rule Them All:
# A Risk-Sensitive Perspective on Model-Based Offline Reinforcement Learning

**Marc Rigter, Bruno Lacerda, Nick Hawes**
Oxford Robotics Institute
University of Oxford
Correspondence: marcrigter@gmail.com

## Abstract

Offline reinforcement learning (RL) is suitable for safety-critical domains where online exploration is not feasible. In such domains, decision-making should take into consideration the risk of catastrophic outcomes. In other words, decision-making should be *risk-averse*. An additional challenge of offline RL is avoiding *distributional shift*, i.e. ensuring that state-action pairs visited by the policy remain near those in the dataset. Previous offline RL algorithms that consider risk combine offline RL techniques (to avoid distributional shift), with risk-sensitive RL algorithms (to achieve risk-aversion). In this work, we propose risk-aversion as a mechanism to jointly address *both* of these issues. We propose a model-based approach, and use an ensemble of models to estimate epistemic uncertainty, in addition to aleatoric uncertainty. We train a policy that is risk-averse, and avoids high uncertainty actions. Risk-aversion to epistemic uncertainty prevents distributional shift, as areas not covered by the dataset have high epistemic uncertainty. Risk-aversion to aleatoric uncertainty discourages actions that are risky due to environment stochasticity. Thus, by considering epistemic uncertainty via a model ensemble and introducing risk-aversion, our algorithm (1R2R) avoids distributional shift in addition to achieving risk-aversion to aleatoric risk. Our experiments show that 1R2R achieves strong performance on deterministic benchmarks, and outperforms existing approaches for risk-sensitive objectives in stochastic domains.

## 1 Introduction

In safety-critical applications, online reinforcement learning (RL) is impractical due to the requirement for extensive random exploration that may be dangerous or costly. To alleviate this issue, offline RL aims to find the best possible policy using only pre-collected data. In safety-critical settings we often want decision-making to be *risk-averse*, and take into consideration variability in the performance between each episode. However, the vast majority of offline RL research considers the expected value objective, and therefore disregards such variability.

A core issue in offline RL is avoiding distributional shift: to obtain a performant policy, we must ensure that the state-action pairs visited by the policy remain near those covered by the dataset. Another way that we can view this is that we need the policy to be averse to *epistemic* uncertainty (uncertainty stemming from a lack of data). In regions not covered by the dataset, epistemic uncertainty is high and there will be large errors in any learned model or value function. A naively optimised policy may exploit these errors, resulting in erroneous action choices.

In risk-sensitive RL, the focus is typically on finding policies that are risk-averse to environment stochasticity, i.e. *aleatoric* uncertainty. Previous approaches to risk-sensitive offline RL [47, 68]

37th Conference on Neural Information Processing Systems (NeurIPS 2023).

combine two techniques: offline RL techniques, such as conservative value function updates [47] or behaviour cloning constraints [68], to avoid distributional shift; and risk-sensitive RL algorithms, to achieve risk-aversion. Therefore, the issues of epistemic uncertainty and aleatoric uncertainty are handled *separately*, using different techniques. In contrast, we propose to avoid epistemic uncertainty due to distributional shift, as well as aleatoric uncertainty due to environment stochasticity, by computing a policy that is risk-averse to *both* sources of uncertainty. This enables us to obtain a performant and risk-averse offline RL algorithm that is simpler than existing approaches.

In this work we propose 1R2R (Figure 1), a model-based algorithm for risk-averse offline RL. The intuition behind our approach is that the policy should avoid the risk of a bad outcome, regardless of whether this risk is due to high uncertainty in the model (epistemic uncertainty) or due to high stochasticity in the environment (aleatoric uncertainty). To represent epistemic uncertainty, we learn an ensemble of models. To optimise a risk-averse policy, we penalise taking actions that are predicted to have highly variable outcomes. In out-of-distribution regions, the disagreement between members of the ensemble is high, producing high variability in the model predictions. Therefore, the risk-averse policy is trained to avoid out-of-distribution regions. Risk-aversion also ensures that the policy avoids

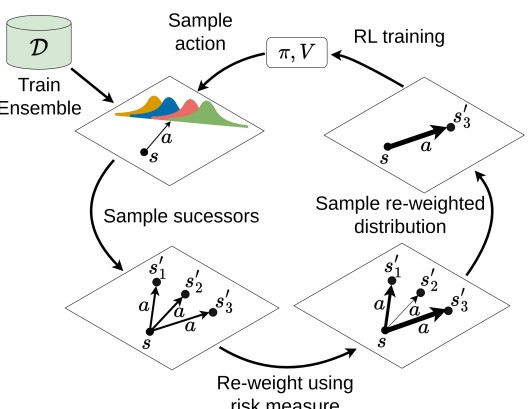

Figure 1: Illustration of the 1R2R algorithm.

risk due to highly stochastic outcomes, i.e. avoids aleatoric risk. Our work demonstrates that incorporating *risk-aversion alone* is a promising approach to offline RL, and proposes an algorithm that is simpler than existing approaches to risk-averse offline RL [47, 68]. Our main contributions are (a) proposing risk-aversion towards epistemic uncertainty as a mechanism to address distributional shift in offline RL, and (b) introducing the first model-based algorithm for risk-averse offline RL, which jointly addresses epistemic and aleatoric uncertainty.

We evaluate our approach in several scenarios. Our experiments show that our algorithm achieves strong performance relative to state-of-the-art baselines on deterministic benchmarks [21], and outperforms existing approaches for risk-sensitive objectives in stochastic domains.

## 2 Related Work

**Offline RL:** Offline RL addresses the problem of learning policies from fixed datasets. Most works on offline RL are *risk-neutral*, meaning that they optimise the expected value [53]. A key issue in offline RL is avoiding distributional shift so that the states visited by the learnt policy remain within the distribution covered by the dataset [42]. Approaches to model-free offline RL include: constraining the learnt policy to be similar to the behaviour policy [23, 38, 71, 22], conservative value function updates [11, 37, 40, 72], importance sampling algorithms [43, 51], or using Q-ensembles for more robust value estimates [1, 3, 73].

Model-based approaches learn a model of the environment and generate synthetic data from that model [63] to optimise a policy via planning [5] or RL algorithms [76, 75]. By training a policy on additional synthetic data, model-based approaches have the potential for broader generalisation [7]. Approaches to preventing model exploitation include constraining the trained policy to be similar to the behaviour policy [64] or applying conservative value function updates [75] in the same fashion as model-free approaches. Another approach is to modify the learnt MDP by either applying reward penalties for state-action pairs with high uncertainty [35, 44, 74, 76], or modifying the transition model to produce pessimistic synthetic transitions [29, 58]. Our work is similar in spirit to [29, 58], but these existing works do not consider risk.

**Risk-Sensitive RL:** Many works have argued for risk-sensitive algorithms in safety-critical domains [25, 48]. Risk-sensitive algorithms for MDPs include approaches that adapt policy gradients to risk objectives [12, 66, 65, 67]. More recently, a number of works have utilised distributional RL [9, 50], which learns a distributional value function that predicts the full return distribution. Knowledge of the return distribution can then be used to optimise a risk-sensitive criterion [15, 33, 46].

The main focus of research on risk-sensitive online RL is risk due to environment stochasticity [19] (i.e. aleatoric uncertainty). However, there are a number of works that explicitly address risk due to epistemic uncertainty [14, 16, 19, 18, 57]. To our knowledge, this is the first work to consider risk-aversion to epistemic uncertainty as a means to mitigate distributional shift in offline RL.

**Risk-Sensitive Offline RL:** To our knowledge, there are two previous algorithms for risk-sensitive offline RL: ORAAC [68] and CODAC [47]. To avoid distributional shift, ORAAC utilises policy constraints, while CODAC employs pessimistic value function updates to penalise out-of-distribution state-action pairs. Both algorithms learn a distributional value function [9] which is then used to optimise a risk-averse policy. Thus, both of these works are model-free, and combine techniques from offline RL and distributional RL to address the issues of distributional shift and risk-sensitivity, respectively. In this work, we provide a new perspective by proposing to optimise for risk-aversion towards *both* epistemic uncertainty (stemming from distributional shift) and aleatoric uncertainty (stemming from environment stochasticity). Unlike previous approaches, our approach (a) is conceptually simpler as it does not combine different techniques, (b) demonstrates that risk-aversion alone is sufficient to avoid distributional shift, (c) does not require a distributional value function (which might be computationally demanding to train), and (d) is model-based.

## 3 Preliminaries

An MDP is defined by the tuple $M = (S, A, T, R, s_0, \gamma)$. $S$ and $A$ are the state and action spaces, $R(s, a)$ is the reward function, $T(s'|s, a)$ is the transition function, $s_0$ is the initial state, and $\gamma \in (0, 1)$ is the discount factor. In this work we consider Markovian policies, $\pi \in \Pi$, which map each state to a distribution over actions. In offline RL we only have access to a fixed dataset of transitions from the MDP: $\mathcal{D} = \{(s_i, a_i, r_i, s_i')\}_{i=1}^{|\mathcal{D}|}$. The goal is to find a performant policy using the fixed dataset.

**Model-Based Offline RL** These approaches utilise a model of the MDP. The learnt dynamics model, $\widehat{T}$, is typically trained via maximum likelihood estimation: $\min_{\widehat{T}} \mathbb{E}_{(s,a,s') \sim \mathcal{D}} \left[ -\log \widehat{T}(s'|s, a) \right]$. A model of the reward function, $\widehat{R}(s, a)$, is also learnt if it is unknown. This results in the learnt MDP model: $\widehat{M} = (S, A, \widehat{T}, \widehat{R}, s_0, \gamma)$. Thereafter, any planning or RL algorithm can be used to recover the optimal policy in the learnt model, $\widehat{\pi} = \arg\max_{\pi \in \Pi} J_{\widehat{M}}^{\pi}$. Here, $J$ indicates the optimisation objective, which is typically the expected value of the discounted cumulative reward.

However, directly applying this approach does not perform well in the offline setting due to distributional shift. In particular, if the dataset does not cover the entire state-action space, the model will inevitably be inaccurate for some state-action pairs. Thus, naive policy optimisation on a learnt model in the offline setting can result in *model exploitation* [32, 41, 55], i.e. a policy that chooses actions that the model erroneously predicts will lead to high reward. We propose to mitigate this issue by learning an ensemble of models and optimising a risk-averse policy.

Following previous works [35, 75, 76], our approach utilises model-based policy optimisation (MBPO) [32]. MBPO performs standard off-policy RL using an augmented dataset $\mathcal{D} \cup \widehat{\mathcal{D}}$, where $\widehat{\mathcal{D}}$ is synthetic data generated by simulating short rollouts in the learnt model. To train the policy, minibatches of data are drawn from $\mathcal{D} \cup \widehat{\mathcal{D}}$, where each datapoint is sampled from the real data, $\mathcal{D}$, with probability $f$, and from $\widehat{\mathcal{D}}$ with probability $1 - f$.

**Risk Measures** We denote the set of all probability distributions by $\mathcal{P}$. Consider a probability space, $(\Omega, \mathcal{F}, P)$, where $\Omega$ is the sample space, $\mathcal{F}$ is the event space, and $P \in \mathcal{P}$ is a probability distribution over $\mathcal{F}$. Let $\mathcal{Z}$ be the set of all random variables defined over the probability space $(\Omega, \mathcal{F}, P)$. We will interpret a random variable $Z \in \mathcal{Z}$ as a reward, i.e. the greater the realisation of $Z$, the better. We denote a $\xi$-weighted expectation of $Z$ by $\mathbb{E}_{\xi}[Z] := \int_{\omega \in \Omega} P(\omega)\xi(\omega)Z(\omega)\mathrm{d}\omega$, where $\xi : \Omega \to \mathbb{R}$ is a function that re-weights the original distribution of $Z$.

A *risk measure* is a function, $\rho : \mathcal{Z} \to \mathbb{R}$, that maps any random variable $Z$ to a real number. An important class of risk measures are *coherent* risk measures, which satisfy a set of properties that are consistent with rational risk assessments. A detailed description and motivation for coherent risk measures is given by [48]. An example of a coherent risk measure is the conditional value at risk (CVaR). CVaR at confidence level $\alpha$ is the mean of the $\alpha$-portion of the worst outcomes.

All coherent risk measures can be represented using their dual representation [6, 61]. A risk measure, $\rho$, is coherent if and only if there exists a convex bounded and closed set, $\mathcal{B}_\rho \in \mathcal{P}$, such that

$$\rho(Z) = \min_{\xi \in \mathcal{B}_\rho(P)} \mathbb{E}_\xi[Z] \tag{1}$$

Thus, the value of any coherent risk measure for any $Z \in \mathcal{Z}$ can be viewed as an $\xi$-weighted expectation of $Z$, where $\xi$ is the worst-case weighting function chosen from a suitably defined *risk envelope*, $\mathcal{B}_\rho(P)$. For more details on coherent risk measures, we refer the reader to [61].

**Static and Dynamic Risk**   In MDPs, the random variable of interest is usually the total reward received at each episode, i.e. $Z_{\mathrm{MDP}} = \sum_{t=0}^{\infty} \gamma^t R(s_t, a_t)$. The *static* perspective on risk applies a risk measure directly to this random variable: $\rho(Z_{\mathrm{MDP}})$. Thus, static risk does not take into account the temporal structure of the random variable in sequential problems, such as MDPs. This leads to the issue of *time inconsistency* [10], meaning that actions which are considered less risky at one point in time may not be considered less risky at other points in time. This leads to non-Markovian optimal policies, and prohibits the straightforward application of dynamic programming due to the coupling of risk preferences over time.

This motivates *dynamic* risk measures, which take into consideration the sequential nature of the stochastic outcome, and are therefore time-consistent. Markov risk measures [60] are a class of dynamic risk measures which are obtained by recursively applying static coherent risk measures. Throughout this paper we will consider dynamic Markov risk over an infinite horizon, denoted by $\rho_\infty$. For policy $\pi$, MDP $M$, and static risk measure $\rho$, the Markov coherent risk $\rho_\infty^\pi(M)$ is defined as

$$\rho_\infty^\pi(M) = R(s_0, \pi(s_0)) + \rho\Big(\gamma R(s_1, \pi(s_1)) + \rho\big(\gamma^2 R(s_2, \pi(s_2)) + \dots\big)\Big), \tag{2}$$

where $\rho$ is a static coherent risk measure, and $(s_0, s_1, s_2, \dots)$ indicates random trajectories drawn from the Markov chain induced by $\pi$ in $M$. Note that in Equation 2 the static risk measure is evaluated at each step according to the distribution over possible successor states.

We define the risk-sensitive value function under some policy $\pi$ as $V^\pi(s, M) = \rho_\infty^\pi(M|s_0 = s)$. This value function can be found by recursively applying the *risk-sensitive* Bellman equation [60]:

$$V^\pi(s, M) = R(s, \pi(s)) + \gamma \min_{\xi \in \mathcal{B}_\rho(T(s, \pi(s), \cdot))} \sum_{s'} T(s, \pi(s), s') \cdot \xi(s') \cdot V^\pi(s', M). \tag{3}$$

Likewise, the optimal risk-sensitive value, $V^*(s, M) = \max_{\pi \in \Pi} \rho_\infty(M|s_0 = s)$, is the unique solution to the risk-sensitive Bellman optimality equation:

$$V^*(s, M) = \max_{a \in A} \Big\{ R(s, a) + \gamma \min_{\xi \in \mathcal{B}_\rho(T(s, a, \cdot))} \sum_{s'} T(s, a, s') \cdot \xi(s') \cdot V^*(s', M) \Big\}. \tag{4}$$

Thus, we can view Markov dynamic risk measures as the expected value in an MDP where the transition probabilities at *each step* are modified *adversarially* within the risk-envelope for the chosen one-step static risk measure.

# 4   One Risk to Rule Them All

In this section, we present *One Risk to Rule Them All* (1R2R), a new algorithm for risk-averse offline RL (Figure 1). Our approach assumes that we can compute an approximation to the posterior *distribution over MDPs*, given the offline dataset. This distribution represents the epistemic uncertainty over the real environment, and enables us to reason about risk due to epistemic uncertainty. 1R2R then uses an RL algorithm to train a policy offline using synthetic data generated from the distribution over MDP models. To achieve risk-aversion, the transition distribution of the model rollouts is modified adversarially according to the risk envelope of a chosen risk measure. This simple modification to standard model-based RL penalises high uncertainty actions, thus penalising actions that have high epistemic uncertainty (i.e. are out-of-distribution) as well as those that have high aleatoric uncertainty (i.e. are inherently risky). Therefore, by modifying the transition distribution of model rollouts to induce risk-aversion, 1R2R avoids distributional shift *and* generates risk-averse behaviour.

## 4.1   Problem Formulation

To simplify the notation required, we assume that the reward function is known and therefore it is only the transition function that must be learnt. Note that in practice, we also learn the reward function

from the dataset. Given the offline dataset, $\mathcal{D}$, we assume that we can approximate the posterior distribution over the MDP transition function given the dataset: $P(T \mid \mathcal{D})$. We can integrate over all possible transition functions to obtain a single expected transition function:

$$\overline{T}(s, a, s') = \int_T T(s, a, s') \cdot P(T \mid \mathcal{D}) \cdot \mathrm{d}T. \tag{5}$$

This gives rise to a single MDP representing the distribution over plausible MDPs, $\overline{M} = (S, A, \overline{T}, R, s_0, \gamma)$. This is similar to the *Bayes-Adaptive* MDP [27, 17, 28] (BAMDP), except that in the BAMDP the posterior distribution over transition functions is updated after each step of online interaction with the environment. Here, we assume that the posterior distribution is fixed given the offline dataset. To make this distinction, we refer to $\overline{M}$ as the Bayesian MDP (BMDP) [4].

In this work, we pose offline RL as optimising the dynamic risk in the BMDP induced by the dataset.

**Problem 4.1** (Risk-Averse Bayesian MDP for Offline RL). Given offline dataset, $\mathcal{D}$, static risk measure $\rho$, and belief over the transition dynamics given the dataset, $P(T \mid \mathcal{D})$, find the policy with the optimal dynamic risk in the corresponding Bayesian MDP $\overline{M}$:

$$\pi^* = \arg\max_\pi \rho_\infty^\pi(\overline{M}) \tag{6}$$

## 4.2 Motivation

The motivation for Problem 4.1 is that optimising for risk-aversion in the Bayesian MDP results in risk-aversion to both aleatoric and epistemic uncertainty. We first provide intuition for this approach via the following empirical example. Then, we demonstrate this theoretically for the case of Gaussian transition models in Proposition 4.2.

**Illustrative Example** Figure 2 illustrates a dataset for an MDP where each episode consists of one step. A single reward is received, equal to the value of the successor state, $s'$. We approximate $P(T|\mathcal{D})$ using an ensemble of neural networks, and the shaded region represents the aggregated Bayesian MDP transition function $\overline{T}(s, a, s')$.

The dashed red line indicates the standard Q-values computed by drawing transitions from $\overline{T}(s, a, s')$. For this value function, the action with the highest Q-value is $a = 1$, which is outside of the data distribution. The solid red line indicates the *risk-sensitive* value function for the risk measure $\text{CVaR}_{0.1}$ (i.e. $\alpha = 0.1$). This is computed by drawing transitions from a uniform distribution over the worst 10% of transitions from $\overline{T}(s, a, s')$. We observe that the risk-sensitive value function penalises straying far from the dataset, where epistemic uncertainty is high. The action in the dataset with highest expected value is $a = 0.4$. However, the action $a = -0.35$ has a higher risk-sensitive value than $a = 0.4$, due to the latter having high-variance transitions (i.e. high aleatoric uncertainty). Thus, we observe that choosing actions according to the risk-sensitive value function penalises actions that are out-of-distribution (i.e. have high epistemic uncertainty) *or* have high aleatoric uncertainty.

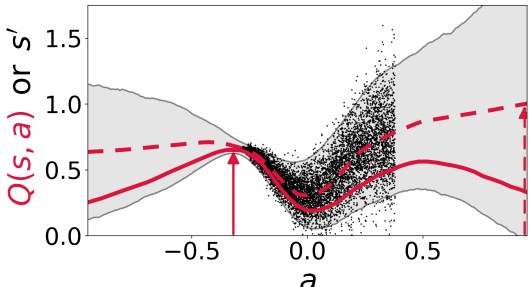

Figure 2: MDP with a single transition to a successor state $s'$, with reward $R(s') = s'$. Black dots illustrate dataset. Shaded region indicates $\pm 2$ S.D. of successor states drawn from an approximation of $\overline{T}(s, a, \cdot)$ (represented by an ensemble [13]). Dashed red line indicates the standard Q-values from running MBPO [32]. Solid red line indicates the risk-sensitive value function computed by reweighting the transitions according to $\text{CVaR}_{0.1}$. Vertical arrows indicate optimal actions for each Q-function.

We now consider the specific case where each member of the ensemble is a Gaussian transition function with standard deviation $\sigma_A$. We assume that in the ensemble, the mean of each Gaussian is also normally distributed with standard deviation $\sigma_E$. The latter assumption allows us to quantify the epistemic uncertainty in terms of the level of disagreement between members of the ensemble. Proposition 4.2 shows that risk-averse optimisation of transitions sampled from the ensemble results in risk-aversion to both the aleatoric and epistemic uncertainty.

**Proposition 4.2.** *Consider some state $s$ in a 1D state space, and some action $a$. Assume (a) that there is an ensemble of $N$ Gaussian transition functions, each denoted $T_i$ with mean $\mu_i$ and standard deviation $\sigma_A$: $\{T_i(s'|s,a) = \mathcal{N}(\mu_i, \sigma_A^2)\}_{i=1}^N$; and (b) that the mean of each Gaussian, $\mu_i$, is normally distributed with mean $\mu_0$ and standard deviation $\sigma_E$: $\mu_i \sim \mathcal{N}(\mu_0, \sigma_E^2)$. The $N$ transition functions jointly define $\overline{T}$, the transition function for Bayesian MDP, $\overline{M}$. Assume that for some policy $\pi$, the risk-sensitive value is linear around $\mu_0$ with some linearity constant, $K$:*

$$V^\pi(s', \overline{M}) = V^\pi(\mu_0, \overline{M}) + K(s' - \mu_0)$$

*Then, the value of a randomly drawn successor state is distributed according to* $\mathcal{N}\Big(\mu = V^\pi(\mu_0, \overline{M}), \sigma^2 = K^2(\sigma_A^2 + \sigma_E^2)\Big)$.

In Proposition 4.2, $\sigma_E$ defines the level of disagreement between the members of the ensemble, and therefore represents the level of epistemic uncertainty. $\sigma_A$ represents the aleatoric uncertainty. The proposition shows that if either $\sigma_A$ or $\sigma_E$ are high, then there is high variability over the value of the successor state when sampling from the ensemble (i.e. sampling from $\overline{T}$ in Equation 5). Risk measures penalise high variability. Therefore, applying the risk-sensitive Bellman equation (Equation 3) to samples from $\overline{T}$ penalises executing state-action pairs for which either $\sigma_A$ or $\sigma_E$ is high. For the specific case of CVaR, Corollary 4.3 illustrates how the risk-sensitive value is reduced as $\sigma_A$ or $\sigma_E$ increase.

**Corollary 4.3.** *Under the assumptions in Proposition 1, the CVaR at confidence level $\alpha$ of the value of a randomly drawn successor state is:*

$$\mathrm{CVaR}_\alpha\big(V^\pi(s', \overline{M}) \mid s' \sim \overline{T}(\cdot \mid s, a)\big) = V^\pi(\mu_0, \overline{M}) - \frac{|K|\sqrt{\sigma_E^2 + \sigma_A^2}}{\alpha\sqrt{2\pi}} \exp^{-\frac{1}{2}(\Phi^{-1}(\alpha))^2}$$

*where $\Phi^{-1}$ is the inverse of the standard normal CDF.*

These results demonstrate that optimising for risk-aversion in the Bayesian MDP (Problem 4.1) results in risk-aversion to both aleatoric and epistemic uncertainty, as state-action pairs with *either* high epistemic or high aleatoric uncertainty are penalised. Therefore, Problem 4.1 favours choosing actions that have both low aleatoric and low epistemic uncertainty.

## 4.3 Approach

Our algorithm is inspired by the model-free online RL method for *aleatoric risk only* introduced by [65], but we have adapted it to the model-based offline RL setting. Following previous works on model-based offline RL, our algorithm makes use of a standard actor-critic off-policy RL algorithm for policy optimisation [58, 75, 76]. However, the key idea of our approach is that to achieve risk-aversion, we modify the distribution of trajectories sampled from the model ensemble. Following Equation 3, we first compute the adversarial perturbation to the transition distribution:

$$\xi^* = \underset{\xi \in \mathcal{B}_\rho(\overline{T}(s,a,\cdot))}{\arg\min} \sum_{s'} \overline{T}(s, a, s') \cdot \xi(s') \cdot V^\pi(s', \overline{M}), \tag{7}$$

where the value function $V^\pi$ is estimated by the off-policy RL algorithm. When generating synthetic rollouts from the model, for each state-action pair $(s, a)$ we sample successor states from the perturbed distribution $\xi^*\overline{T}$:

$$s' \sim \xi^*(s') \cdot \overline{T}(s, a, s') \ \forall s' \tag{8}$$

This perturbed distribution increases the likelihood of transitions to low-value states. The transition data sampled from this perturbed distribution is added to the synthetic dataset, $\widehat{\mathcal{D}}$. This approach modifies the sampling distribution over successor states according to the risk-sensitive Bellman equation in Equation 3. Therefore, the value function learnt by the actor-critic RL algorithm under this modified sampling distribution is equal to the risk-sensitive value function. The policy is then optimised to maximise the risk-sensitive value function.

For the continuous problems we address, we cannot compute the adversarially modified transition distribution in Equation 7 exactly. Therefore, we use a sample average approximation (SAA) [62] to approximate the solution. Specifically, we approximate $\overline{T}(s, a, \cdot)$ as a uniform distribution over $m$ states drawn from $\overline{T}(s, a, \cdot)$. Then, we compute the optimal adversarial perturbation to this uniform distribution over successor states. Under standard regularity assumptions, the solution to the SAA

---

**Algorithm 1** 1R2R

---

1: **Input:** Offline dataset, $\mathcal{D}$; Static risk measure, $\rho$;
2: Compute belief distribution over transition models, $P(T \mid \mathcal{D})$;
3: **for** iter $= 1, \ldots, N_{\text{iter}}$ :
4:     **for** rollout $= 1, \ldots, N_{\text{rollout}}$, generate synthetic rollouts:
5:         Sample initial state, $s \in \mathcal{D}$.
6:         **for** step $= 1, \ldots, k$ :
7:             Sample action $a$ according to current policy $\pi(s)$.
8:             Sample $m$ successor states, $\Psi = \{s'_i\}_{i=1}^m$ from $\overline{T}(s, a, \cdot)$.
9:             Consider $\widehat{T}$, a discrete approximation to $\overline{T}$:   $\widehat{T}(s, a, s') \approx \overline{T}(s, a, s') = \frac{1}{m}, \; \forall s' \in \Psi$
10:            Compute adversarial perturbation to $\widehat{T}$:

$$\widehat{\xi}^* = \underset{\xi \in \mathcal{B}_\rho(\widehat{T}(s,a,\cdot))}{\arg\min} \sum_{s' \in \Psi} \widehat{T}(s, a, s') \cdot \xi(s') \cdot V^\pi(s', \overline{M})$$

11:            Sample from adversarially modified distribution: $s' \sim \widehat{\xi}^* \cdot \widehat{T}(s, a, \cdot)$
12:            Add $(s, a, R(s, a), s')$ to the synthetic dataset, $\widehat{\mathcal{D}}$.
13:            $s \leftarrow s'$
14:     *Agent update*: Update $\pi$ and $V^\pi$ with an actor-critic algorithm, using samples from $\mathcal{D} \cup \widehat{\mathcal{D}}$.

---

converges to the exact solution as $m \to \infty$ [62]. Details of how the perturbed distribution is computed for CVaR and the Wang risk measure are in Appendix B.

**Algorithm** Our approach is summarised in Algorithm 1 and illustrated in Figure 1. At each iteration, we generate $N_{\text{rollout}}$ truncated synthetic rollouts which are branched from an initial state sampled from the dataset (Lines 4-6). For a given $(s, a)$ pair we sample a set $\Psi$ of $m$ candidate successor states from $\overline{T}$ (Line 8). We then approximate the original transition distribution by $\widehat{T}$, a uniform distribution over $\Psi$, in Line 9. Under this approximation, it is straightforward to compute the worst-case perturbed transition distribution for a given risk measure in Line 10 (see Appendix B). The final successor state $s'$ is sampled from the perturbed distribution in Line 11, and the transition to this final successor state is added to $\widehat{\mathcal{D}}$.

After generating the synthetic rollouts, the policy and value function are updated using an off-policy actor-critic algorithm by sampling data from both the synthetic dataset and the real dataset (Line 14). Utilising samples from the real dataset means that Algorithm 1 does not precisely replicate the risk-sensitive Bellman equation (Equation 3). However, like [58, 75] we found that additionally utilising the real data improves performance (see Appendix D.4).

**Implementation Details** Following a number of previous works [29, 54, 76] we use a uniform distribution over an ensemble of neural networks to approximate $P(T \mid \mathcal{D})$. We also learn an ensemble of reward functions from the dataset, in addition to the transition model. Each model in the ensemble, $T_i$, outputs a Gaussian distribution over successor states and rewards: $T_i(s', r \mid s, a) = \mathcal{N}(\mu_i(s, a), \Sigma_i(s, a))$. For policy training we used soft-actor critic (SAC) [30] in Line 14. For policy improvement, we used the standard policy gradient that is utilised by SAC. We found this worked well in practice and meant that our approach requires minimal modifications to existing RL algorithms. Research on specialised policy gradients for risk-sensitive objectives can be found in [65, 12].

## 5 Experiments

In our experiments, we seek to: (a) verify that 1R2R generates performant policies in deterministic environments by avoiding distributional shift, (b) investigate whether 1R2R generates risk-averse behaviour on stochastic domains, and (c) compare the performance of 1R2R against existing baselines on both stochastic and deterministic environments. To examine each of these questions, we evaluate our approach on the following domains. More information about the domains can be found in Appendix E.2, and code for the experiments can be found at github.com/marc-rigter/1R2R.

**D4RL MuJoCo** [21] There are three deterministic robot environments (*HalfCheetah, Hopper, Walker2D*), each with 4 datasets (*Random, Medium, Medium-Replay, Medium-Expert*).

**Currency Exchange** We adapt the Optimal Liquidation problem [2] to offline RL. The agent aims to convert 100 units of currency A to currency B, subject to a stochastic exchange rate, before a deadline. The dataset consists of experience from a random policy.

**HIV Treatment** We adapt this online RL domain [33, 20] to the offline setting. The 6-dimensional continuous state vector represents the concentrations of various cells and viruses. The actions correspond to the dosages of two drugs. Transitions are stochastic due to the efficacy of each drug varying stochastically at each step. The reward is determined by the health outcomes for the patient, and penalises side-effects due to the quantity of drugs prescribed. The dataset is constructed in the same way as the D4RL [21] *Medium-Replay* datasets.

**Stochastic MuJoCo** We modify the D4RL benchmarks by adding stochastic perturbation forces. We focus on Hopper and Walker2D as there is a risk that any episode may terminate early due to the robot falling over. We vary the magnitude of the perturbations between two different levels (*Moderate-Noise, High-Noise*). For each domain and noise level we construct *Medium, Medium-Replay*, and *Medium-Expert* datasets in the same fashion as D4RL.

**Baselines** We compare 1R2R against offline RL algorithms designed for risk-sensitivity (ORAAC [68] and CODAC [47]) in addition to performant risk-neutral model-based (RAMBO [58], COMBO [75], and MOPO [76]) and model-free (IQL [38], TD3+BC [22], CQL [40], and ATAC [11]) algorithms. Due to computational constraints, we only compare a subset of algorithms for Stochastic MuJoCo. Following [21], the results are normalised to approximately between 0 and 100.

**Objectives and Hyperparameters** A disadvantage of the dynamic risk perspective used by 1R2R is that it is unclear how to evaluate performance for dynamic risk. For this reason, we evaluate the algorithms using static risk measures. For D4RL, which is deterministic, we optimise the standard expected value objective. Following [68, 47], we use static $\text{CVaR}_{0.1}$ (CVaR at confidence level $\alpha = 0.1$) as the optimisation objective for all other domains. This is the average total reward received on the worst 10% of runs. For ORAAC and CODAC, we set the optimisation objective to the desired objective for each domain. For all other baselines, we tune hyperparameters to obtain the best online performance for the desired objective on each dataset (see Appendix E.5). Likewise, we tune the hyperparameters for 1R2R (the rollout length and risk measure parameter) to achieve the best online performance as described in Appendix C.4.

**Versions of 1R2R** We test two versions of 1R2R which utilise different risk measures to re-weight the transition distribution to successor states: $\text{1R2R}_{\text{CVaR}}$ and $\text{1R2R}_{\text{Wang}}$. These risk measures are defined formally in Appendix B. We chose CVaR because it is commonly used and easy to interpret. However, CVaR disregards the best possible outcomes, and this can lead to an unnecessary loss in performance [56]. Therefore, we also investigate using the Wang risk measure which is computed by reweighting the entire distribution over outcomes.

We present results for two ablations. *Ablate risk* is the same as 1R2R, except that the transition distribution of the rollouts is not modified, making this ablation equivalent to standard model-based RL. *Ablate ensemble* uses only a single model, and therefore the risk aversion is only with respect to the aleatoric uncertainty.

**Results Presentation** The results in Tables 1-3 are from evaluating the policies over the last 10 iterations of training, and averaged over 5 seeds. The highlighted numbers indicate results within 10% of the best score, $\pm$ indicates the standard deviation over seeds, and "div." indicates that the value function diverged for at least one training run.

## 5.1 Results

**D4RL MuJoCo** For these domains, the objective is to optimise the expected performance. Therefore, to generate the results in Table 1 we tune all algorithms to obtain the best expected performance. Note that to obtain strong expected performance on these domains, 1R2R still utilises risk-aversion to avoid distributional shift. Table 1 shows that both configurations of 1R2R outperform most of the baselines, and are competitive with the strongest risk-neutral baselines. The two existing offline RL algorithms that are able to consider risk, ORAAC and CODAC, perform poorly on these benchmarks. The ablation of the risk-sensitive sampling shows that when risk-sensitivity is removed from 1R2R, for many domains the training diverges (denoted by "div.") or performance degrades. This verifies that risk-aversion is crucial in our approach to ensure that the policy avoids distributional shift, and therefore also avoids the associated problem of value function instability [39]. Likewise, the ablation of the model ensemble also leads to unstable training for some datasets, indicating that risk-aversion to epistemic uncertainty (represented by the ensemble) is crucial to our approach.

**Currency Exchange** From now on, we consider stochastic domains where the objective is to obtain the best performance for the static $\text{CVaR}_{0.1}$ objective. All algorithms are tuned to optimise this

Table 1: Normalised expected value results for D4RL MuJoCo-v2.

| | | 1R2R$_{CVaR}$ | 1R2R$_{Wang}$ | Ablate Risk | Ablate Ens. | ORAAC | CODAC | RAMBO | CQL | IQL | TD3+BC | MOPO | ATAC | COMBO |
|---|---|---|---|---|---|---|---|---|---|---|---|---|---|---|
| Random | HalfCheetah | 36.9 ± 6.4 | 35.5 ± 1.7 | 38.6 ± 1.9 | 35.3 ± 4.1 | 22.4 ± 1.7 | 31.0 ± 2.0 | 40.0 | 19.6 | 1.9 | 11.0 | 35.4 | 3.9 | 38.8 |
| | Hopper | 30.9 ± 2.3 | 31.8 ± 0.2 | div. | div. | 10.9 ± 13.3 | 9.2 ± 0.7 | 21.6 | 6.7 | 1.2 | 8.5 | 11.7 | 17.5 | 17.9 |
| | Walker2D | 7.6 ± 12.2 | 8.1 ± 11.0 | div. | div. | 2.4 ± 3.5 | 12.8 ± 8.9 | 11.5 | 2.4 | 0.3 | 1.6 | 13.6 | 6.8 | 7.0 |
| Medium | HalfCheetah | 74.5 ± 2.1 | 75.5 ± 1.2 | 76.9 ± 1.2 | 60.3 ± 18.7 | 38.5 ± 4.8 | 62.8 ± 0.9 | 77.6 | 49.0 | 47.4 | 48.3 | 42.3 | 53.3 | 54.2 |
| | Hopper | 80.2 ± 15.8 | 64.6 ± 25.1 | 64.4 ± 18.5 | 58.1 ± 23.2 | 29.6 ± 10.3 | 63.9 ± 1.7 | 92.8 | 66.6 | 66.3 | 59.3 | 28.0 | 85.6 | 97.2 |
| | Walker2D | 63.9 ± 31.8 | 83.4 ± 4.8 | 85.3 ± 3.6 | 78.2 ± 4.0 | 45.5 ± 14.9 | 84.0 ± 4.0 | 86.9 | 83.8 | 78.3 | 83.7 | 17.8 | 89.6 | 81.9 |
| Medium Replay | HalfCheetah | 65.7 ± 2.3 | 65.6 ± 1.3 | 65.9 ± 3.4 | 62.2 ± 2.8 | 39.3 ± 1.7 | 53.4 ± 1.9 | 68.9 | 47.1 | 44.2 | 44.6 | 53.1 | 48.0 | 55.1 |
| | Hopper | 92.9 ± 10.7 | 93.2 ± 8.7 | 55.3 ± 14.4 | 85.0 ± 27.9 | 22.6 ± 12.1 | 68.8 ± 28.2 | 96.6 | 97.0 | 94.7 | 60.9 | 67.5 | 102.5 | 89.5 |
| | Walker2D | 92.2 ± 2.3 | 88.4 ± 5.0 | 92.7 ± 3.6 | 85.9 ± 11.3 | 11.1 ± 5.6 | 73.9 ± 31.7 | 85.0 | 88.2 | 73.9 | 81.8 | 39.0 | 92.5 | 56.0 |
| Medium Expert | HalfCheetah | 96.0 ± 6.0 | 94.5 ± 1.9 | 87.8 ± 6.3 | 73.0 ± 14.5 | 24.4 ± 7.6 | 76.7 ± 4.9 | 93.7 | 90.8 | 86.7 | 90.7 | 63.3 | 94.8 | 90.0 |
| | Hopper | 81.6 ± 22.9 | 89.6 ± 17.8 | 88.6 ± 13.4 | 45.8 ± 20.1 | 4.1 ± 2.8 | 87.6 ± 6.4 | 83.3 | 106.8 | 91.5 | 98.0 | 23.7 | 111.9 | 111.1 |
| | Walker2D | 90.9 ± 5.9 | 78.1 ± 7.9 | div. | 58.2 ± 26.8 | 42.8 ± 9.2 | 112.0 ± 2.3 | 68.3 | 109.4 | 109.6 | 110.1 | 44.6 | 114.2 | 96.1 |

Table 2: Results on Currency Exchange and HIV Treatment domains. The objective is to optimise static $CVaR_{0.1}$. We report static $CVaR_{0.1}$ in addition to the average performance.

| | 1R2R$_{CVaR}$ | 1R2R$_{Wang}$ | Ablate Risk | Ablate Ens. | ORAAC | CODAC | RAMBO | CQL | IQL | TD3+BC | MOPO | ATAC | COMBO |
|---|---|---|---|---|---|---|---|---|---|---|---|---|---|
| Currency: $CVaR_{0.1}$ | 64.0 ± 1.9 | 62.8 ± 2.3 | 31.4 ± 4.6 | 61.3 ± 3.0 | 0.0 ± 0.0 | 67.6 ± 0.4 | 28.7 ± 0.8 | 41.7 ± 1.3 | 0.0 ± 0.0 | 27.0 ± 8.3 | 55.0 ± 22.2 | 16.9 ± 6.5 | 49.0 ± 25.9 |
| Currency: Average | 78.8 ± 1.6 | 81.6 ± 1.1 | 99.7 ± 1.3 | 80.1 ± 1.9 | 0.0 ± 0.0 | 74.0 ± 0.1 | 99.6 ± 0.9 | 89.4 ± 1.3 | 0.0 ± 0.0 | 100.4 ± 3.6 | 64.7 ± 21.8 | 73.6 ± 8.5 | 55.0 ± 29.0 |
| HIV: $CVaR_{0.1}$ | 53.9 ± 0.5 | 51.4 ± 7.5 | 41.2 ± 21.9 | 47.3 ± 12.4 | 0.5 ± 0.7 | 26.7 ± 0.9 | 52.5 ± 1.8 | 27.2 ± 2.5 | 0.1 ± 0.1 | 35.8 ± 4.0 | 0.0 ± 0.0 | 0.1 ± 0.1 | 11.2 ± 7.3 |
| HIV: Average | 73.5 ± 1.9 | 68.9 ± 10.5 | 56.7 ± 30.1 | 64.3 ± 17.0 | 6.6 ± 13.1 | 59.9 ± 0.5 | 73.3 ± 2.1 | 63.5 ± 1.8 | 27.1 ± 4.9 | 70.5 ± 1.5 | 0.0 ± 0.0 | 1.0 ± 1.7 | 38.7 ± 22.4 |

objective. Table 2 shows the results for the Currency Exchange domain, and a plot of the return distributions is in Appendix D.3. We see that 1R2R and CODAC obtain by far the best performance for static $CVaR_{0.1}$, indicating that these algorithms successfully optimise for risk-aversion. When the risk-averse sampling is ablated, 1R2R performs poorly for $CVaR_{0.1}$ demonstrating that the risk-averse sampling leads to improved risk-sensitive performance. ORAAC and the risk-neutral baselines perform poorly for the static $CVaR_{0.1}$ objective. TD3+BC and RAMBO achieve strong average performance but perform poorly for CVaR, indicating that there is a tradeoff between risk and expected value for this domain.

**HIV Treatment** The results for this domain are in Table 2, and full return distributions are shown in Appendix D.2. For this domain, we observe that 1R2R and RAMBO obtain the best performance for static $CVaR_{0.1}$, as well as for average performance. The existing risk-sensitive algorithms, ORAAC and CODAC, both perform poorly on this domain. A possible explanation for this poor performance is that it is difficult to learn an accurate distributional value function from fixed and highly stochastic datasets. We observe that TD3+BC performs well in expectation, but poorly for static $CVaR_{0.1}$. This shows that strong expected performance is not sufficient to ensure good performance in the worst outcomes.

**Stochastic MuJoCo** Due to resource constraints, we compare a subset of the algorithms on these benchmarks. The performance of each algorithm for static $CVaR_{0.1}$ is in Table 3. We observe that 1R2R$_{CVaR}$ performs the best for static $CVaR_{0.1}$ in the *Medium* and *Medium-Replay* datasets. For the ablation of risk-averse sampling, either training diverges or the performance degrades relative to 1R2R$_{CVaR}$ for most datasets. For these benchmarks, 1R2R$_{Wang}$ performs worse than 1R2R$_{CVaR}$. This may be because the Wang risk measure makes use of the entire distribution over outcomes, and therefore 1R2R$_{Wang}$ trains the policy on some over-optimistic data. On the other hand, 1R2R$_{CVaR}$ disregards the most optimistic outcomes, resulting in more robust performance. Due to these results, we recommend using 1R2R$_{CVaR}$ over 1R2R$_{Wang}$ by default.

1R2R$_{CVaR}$ considerably outperforms the prior risk-sensitive algorithms, ORAAC and CODAC. While IQL performs well for the deterministic benchmarks, it performs very poorly on all stochastic domains. This suggests that we should not rely on deterministic benchmarks for benchmarking offline RL algorithms. Several of the risk-neutral baselines outperform 1R2R$_{CVaR}$ on the multi-modal *Medium-Expert* datasets. This echoes previous findings indicating that model-based approaches perform less well on multi-modal datasets [45]. Thus, while 1R2R achieves very strong performance across all *Random*, *Medium*, and *Medium-Replay* datasets, it may be less suitable for multi-modal datasets. Finally, the expected value results in Table 6 in Appendix D.1 are similar, with the exception that 1R2R$_{CVaR}$ performs slightly less well for expected performance relative to the baselines. This indicates that for these domains, there is a modest tradeoff between risk and expected performance.

## 6 Discussion and Conclusion

The results for the Currency Exchange domain show that only 1R2R and CODAC are able to achieve risk-averse behaviour. On most other domains, 1R2R comfortably outperforms CODAC, in addition

Table 3: Normalised static $\text{CVaR}_{0.1}$ results for Stochastic MuJoCo.

| | | $\text{1R2R}_{\text{CVaR}}$ | $\text{1R2R}_{\text{Wang}}$ | Ablate Risk | ORAAC | CODAC | RAMBO | CQL | IQL | TD3+BC |
|---|---|---|---|---|---|---|---|---|---|---|
| **Medium** | Hopper (Mod) | $36.9\pm5.1$ | $37.3\pm1.9$ | $35.7\pm2.9$ | $10.2\pm3.9$ | $33.6\pm0.1$ | $37.8\pm8.3$ | $33.5\pm0.1$ | $33.0\pm1.4$ | $33.4\pm0.1$ |
| | Walker2D (Mod) | $29.9\pm8.7$ | $28.3\pm4.2$ | $7.7\pm11.6$ | $17.7\pm4.6$ | $26.8\pm0.8$ | $8.0\pm0.5$ | $32.0\pm1.7$ | $22.7\pm2.3$ | $22.9\pm11.5$ |
| | Hopper (High) | $38.6\pm3.0$ | $42.0\pm1.2$ | $40.2\pm1.2$ | $18.5\pm10.0$ | $36.8\pm0.3$ | $36.5\pm7.3$ | $38.7\pm0.8$ | $35.8\pm0.9$ | $37.4\pm0.5$ |
| | Walker2D (High) | $19.4\pm12.6$ | $19.5\pm5.2$ | div. | $6.3\pm7.4$ | $32.2\pm3.6$ | $19.7\pm0.7$ | $43.1\pm2.7$ | $16.7\pm3.5$ | $48.2\pm3.4$ |
| **Medium Replay** | Hopper (Mod) | $43.9\pm11.7$ | $37.1\pm7.6$ | $33.6\pm2.8$ | $7.4\pm1.3$ | $32.1\pm4.7$ | $35.1\pm5.7$ | $34.1\pm2.2$ | $-0.6\pm0.0$ | $24.2\pm2.3$ |
| | Walker2D (Mod) | $30.1\pm7.2$ | $28.0\pm7.1$ | $29.7\pm2.2$ | $4.7\pm2.5$ | $23.3\pm10.4$ | $30.9\pm5.7$ | $25.4\pm1.9$ | $1.4\pm0.6$ | $28.8\pm1.0$ |
| | Hopper (High) | $51.4\pm8.1$ | $46.1\pm7.9$ | $30.5\pm10.9$ | $14.7\pm15.2$ | $38.9\pm0.8$ | $36.4\pm8.6$ | $50.9\pm2.8$ | $7.3\pm5.8$ | $35.5\pm2.0$ |
| | Walker2D (High) | $46.0\pm10.1$ | $28.8\pm5.2$ | $30.9\pm14.3$ | $1.1\pm0.8$ | $43.1\pm14.4$ | $42.2\pm10.4$ | $15.6\pm6.1$ | $-1.2\pm0.3$ | $29.9\pm6.8$ |
| **Medium Expert** | Hopper (Mod) | $65.4\pm31.7$ | $45.7\pm35.7$ | $32.6\pm26.3$ | $14.9\pm9.6$ | $44.1\pm10.9$ | $32.0\pm12.6$ | $60.6\pm11.7$ | $75.0\pm29.7$ | $54.2\pm3.3$ |
| | Walker2D (Mod) | $34.8\pm6.5$ | $38.5\pm8.9$ | div. | $14.7\pm3.6$ | $28.1\pm12.5$ | $29.9\pm14.2$ | $72.3\pm9.0$ | $24.5\pm1.2$ | $68.4\pm2.7$ |
| | Hopper (High) | $35.5\pm5.7$ | $39.3\pm4.6$ | $30.5\pm7.4$ | $9.1\pm9.4$ | $47.1\pm1.7$ | $38.5\pm2.1$ | $47.1\pm2.1$ | $32.6\pm4.3$ | $51.0\pm1.3$ |
| | Walker2D (High) | $10.8\pm4.3$ | $11.7\pm5.2$ | div. | $4.2\pm6.2$ | $26.7\pm16.5$ | $1.9\pm1.4$ | $55.5\pm5.5$ | $28.0\pm8.8$ | $60.2\pm8.9$ |

to most of the baselines. ORAAC, the other existing risk-sensitive algorithm, performs poorly across all domains tested. This demonstrates the advantage of 1R2R over existing algorithms: it is able to generate risk-averse behaviour (unlike the risk-neutral baselines), and it also achieves strong performance across many domains (unlike ORAAC and CODAC). We now discuss the limitations of our approach and possible directions for future work.

**Static vs Dynamic Risk** We chose to base our algorithm on dynamic risk, which evaluates risk recursively at each step, rather than static risk, which evaluates risk across episodes. The dynamic risk perspective is arguably more suitable for avoiding distributional shift, as it enforces aversion to uncertainty at each time step, thus preventing the policy from leaving the data-distribution at any step. It is also more amenable to our model-based approach, where we modify the transition distribution of synthetic data at each step. Applying this same approach to the static risk setting would likely require modifying the distribution over entire trajectories [12, 67]. However, sampling full length trajectories is not desirable in model-based methods due to compounding modelling error [32]. The disadvantage of dynamic risk is that it is difficult to choose the risk measure to apply and it can lead to overly conservative behaviour [24]. Furthermore, it is unclear how to interpret and evaluate performance for dynamic risk. For this reason, we evaluated static risk objectives in our experiments.

**Modelling Stochasticity** In our implementation, the transition distribution for each model is Gaussian. This means that our implementation is not suitable for domains with multimodal transition dynamics. To improve our approach, an obvious next step is to utilise better model architectures that can handle general multimodal distributions, as well as high-dimensional observations [31].

**Epistemic vs Aleatoric Uncertainty** Our approach is agnostic to the source of uncertainty. This is motivated by the intuition that poor outcomes should be avoided due to either source of uncertainty. A direction for future work is to use composite risk measures [18], which enable the aversion to each source of risk to be adjusted independently. This might be important for achieving strong performance if the estimation of each source of uncertainty is poorly calibrated.

**Conclusion** In this paper we proposed 1R2R, a model-based algorithm for risk-sensitive offline RL. Our algorithm is motivated by the intuition that a risk-averse policy should avoid the risk of poor outcomes, regardless of whether that risk is due to a lack of information about the environment (epistemic uncertainty) or high stochasticity in the environment (aleatoric uncertainty). 1R2R is: (a) simpler than existing approaches to risk-sensitive offline RL which combine different techniques, and (b) is able to generate risk-averse behaviour while achieving strong performance across many domains. As discussed, there many promising avenues for future work, including: adapting 1R2R to optimise for static (rather than dynamic) risk; utilising more powerful generative models to model the distribution over MDPs; and using composite risk measures to adjust the aversion to each source of uncertainty independently.

## Acknowledgements

This work was supported by a Programme Grant from the Engineering and Physical Sciences Research Council (EP/V000748/1), the Clarendon Fund at the University of Oxford, and a gift from Amazon Web Services. Additionally, this project made use of time on Tier 2 HPC facility JADE2, funded by the Engineering and Physical Sciences Research Council (EP/T022205/1).

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

# Appendices

## A   Proofs

**Proposition A.1.** *Consider some state $s$ in a 1D state space, and some action $a$. Assume (a) that there is an ensemble of $N$ Gaussian transition functions, each denoted $T_i$ with mean $\mu_i$ and standard deviation $\sigma_A$: $\{T_i(s'|s,a) = \mathcal{N}(\mu_i, \sigma_A^2)\}_{i=1}^{N}$; and (b) that the mean of each Gaussian, $\mu_i$, is normally distributed with mean $\mu_0$ and standard deviation $\sigma_E$: $\mu_i \sim \mathcal{N}(\mu_0, \sigma_E^2)$. The $N$ transition functions jointly define $\overline{T}$, the transition function for Bayesian MDP, $\overline{M}$. Assume that for some policy $\pi$, the risk-sensitive value is linear around $\mu_0$ with some linearity constant, $K$:*

$$V^\pi(s', \overline{M}) = V^\pi(\mu_0, \overline{M}) + K(s' - \mu_0)$$

*Then, the value of a randomly drawn successor state is distributed according to $\mathcal{N}\left(\mu = V^\pi(\mu_0, \overline{M}), \sigma^2 = K^2(\sigma_A^2 + \sigma_E^2)\right)$.*

### A.1   Proof of Proposition 4.2

*Proof*: The joint probability of sampling an ensemble member with mean, $\mu_i$, and then sampling a successor state, $s'$, is:

$$P(\mu_i, s') = P(\mu_i)P(s' \mid \mu_i).$$

We marginalise over $\mu_i$:

$$P(s') = \int_{-\infty}^{\infty} P(\mu_i)P(s' \mid \mu_i)\mathrm{d}\mu_i$$

$P(\mu_i)$ and $P(s' \mid \mu_i)$ are both Gaussian. Substituting the appropriate probability density functions we have that

$$P(s') = \int_{-\infty}^{\infty} \frac{1}{\sigma_E\sqrt{2\pi}} \exp^{-\frac{(\mu_i - \mu_0)^2}{2\sigma_E^2}} \frac{1}{\sigma_A\sqrt{2\pi}} \exp^{-\frac{(s'-\mu_i)^2}{2\sigma_A^2}} \mathrm{d}\mu_i$$

$$= \frac{1}{\sqrt{\sigma_A^2 + \sigma_E^2}\sqrt{2\pi}} \exp^{-\frac{(s'-\mu_0)^2}{2\sqrt{\sigma_E^2+\sigma_A^2}}} = \mathcal{N}(\mu_0, \sigma_E^2 + \sigma_A^2). \quad (9)$$

Thus, $s'$ is normally distributed with mean $\mu_0$ and standard deviation $\sqrt{\sigma_E^2 + \sigma_A^2}$. The random variable $K(s' - \mu_0)$ is therefore normally distributed with mean 0 and standard deviation $|K|\sqrt{\sigma_E^2 + \sigma_A^2}$. Finally, $V(s') = V(\mu_0) + K(s' - \mu_0)$ is also normally distributed with mean $V(\mu_0)$ and standard deviation $|K|\sqrt{\sigma_E^2 + \sigma_A^2}$. $\square$

Corollary 4.3 follows directly from Proposition 1 and the formula for the CVaR for a Gaussian random variable [34].

## B   Risk Measures

In our experiments, we make use of two different risk measures in our algorithm: the conditional value at risk (CVaR) [59], and the Wang risk measure [70]. For parameter $\alpha$, CVaR is the mean of the $\alpha$-portion of the worst outcomes: $\rho_{\text{CVaR}}(Z, \alpha) = \mathbb{E}[z \sim Z \mid z \leq F_Z^{-1}(\alpha)]$, where $F_Z$ is the CDF of $Z$. For parameter $\eta$, the Wang risk measure for random variable $Z$ can be defined as the expectation under a distorted cumulative distribution function $\rho_{\text{Wang}}(Z, \eta) = \int_{-\infty}^{\infty} z \frac{\partial}{\partial z} g(F_Z(z))\mathrm{d}z$, and the distortion function is $g(\tau) = \Phi(\Phi^{-1}(\tau) + \eta)$, where $\Phi$ is the CDF of the standard normal distribution. In both instances, the parameters $\alpha$ and $\eta$ control the level of risk-aversion.

We test our approach with these two risk measures because CVaR is commonly used and easy to interpret, but only considers the $\alpha$-portion of the distribution, while the Wang risk measure takes into consideration the entire distribution. However, the general approach we propose is compatible with any coherent risk measure.

## B.1 Computing Perturbed Distributions

Here, we explain how the distribution over candidate successor states, $s' \in \Psi$ is modified for CVaR and the Wang risk measures in Algorithm 1. To avoid the need for excessive notation we assume that all sampled candidate successor states in $\Psi$ are unique, and that no two successor states have exactly the same value, i.e. $V(s_i') \neq V(s_j')$, for any two different $s_i', s_j' \in \Psi$.

**Conditional Value at Risk**  Consider a probability space $(\Omega, \mathcal{F}, P)$, where $\Omega$ is the sample space, $\mathcal{F}$ is the event space, and $P$ is a probability distribution over $\mathcal{F}$. For a continuous random variable $Z$, CVaR is the mean of the $\alpha$-portion of the worst outcomes: $\rho_{\mathrm{CVaR}}(Z, \alpha) = \mathbb{E}\big[z \sim Z \mid z \leq F_Z^{-1}(\alpha)\big]$, where $F_Z$ is the CDF of $Z$. For CVaR at confidence level $\alpha$, the corresponding risk envelope is:

$$\mathcal{B}_{\mathrm{CVaR}_\alpha}(P) = \left\{ \xi : \xi(\omega) \in \left[0, \frac{1}{\alpha}\right] \forall \omega, \int_{\omega \in \Omega} \xi(\omega) P(\omega) \mathrm{d}\omega = 1 \right\} \tag{10}$$

The risk envelope allows the likelihood for any outcome to be increased by at most $1/\alpha$. In Algorithm 1, for state-action pair $(s, a)$, we approximate the distribution over successor states as a discrete uniform distribution over candidate successor states $s' \in \Psi$, where $\Psi$ is a set of $m$ states sampled from $\overline{T}(s, a, s')$. This discrete distribution is $\widehat{T}(s, a, s') = \frac{1}{m}$ for all $s' \in \Psi$.

In Line 10 of Algorithm 1, we denote by $\widehat{\xi}^*$ the adversarial perturbation to the discrete transition distribution:

$$\widehat{\xi}^* = \underset{\xi \in \mathcal{B}_\rho(\widehat{T}(s,a,\cdot))}{\arg\min} \sum_{s' \in \Psi} \widehat{T}(s, a, s') \cdot \xi(s') \cdot V^\pi(s')$$

Define $\mathcal{V}_{s'}$ as the random variable representing the value of the successor state, $V^\pi(s')$, when the successor state is drawn from $s' \sim \widehat{T}(s, a, \cdot)$. Also define the value at risk at confidence level $\alpha \in (0, 1]$, which is the $\alpha$-quantile of a distribution: $\mathrm{VaR}_\alpha(Z) = \min\{z \mid F_Z(z) \geq \alpha\}$. Then, from the risk envelope in Equation 10, we can derive that the adversarial perturbation is:

$$\widehat{\xi}_{\mathrm{CVaR}}(s') \cdot \widehat{T}(s, a, s') = \begin{cases} \frac{1}{m \cdot \alpha}, & \text{if } V(s') < \mathrm{VaR}_\alpha(\mathcal{V}_{s'}). \\ 0, & \text{if } V(s') > \mathrm{VaR}_\alpha(\mathcal{V}_{s'}). \\ 1 - \sum_{s' \in \Psi} \frac{1}{m \cdot \alpha} \mathbb{1}[V(s') < \mathrm{Var}_\alpha], & \text{if } V(s') = \mathrm{VaR}_\alpha(\mathcal{V}_{s'}). \end{cases} \tag{11}$$

The last line of Equation 11 ensures that the perturbed distribution is a valid probability distribution (as required by the risk envelope in Equation 10).

**Wang Risk Measure**  For parameter $\eta$, the Wang risk measure for random variable, $Z$, can be defined as the expectation under a distorted cumulative distribution function $\rho_{\mathrm{Wang}}(Z, \eta) = \int_{-\infty}^{\infty} z \frac{\partial}{\partial z} g(F_Z(z)) \mathrm{d}z$. The distortion function is $g(\tau) = \Phi(\Phi^{-1}(\tau) + \eta)$, where $\Phi$ is the CDF of the standard normal distribution.

To compute the perturbation to $\widehat{T}(s, a, \cdot)$ according to the Wang risk measure, we order the candidate successor states according to their value. We use the subscript to indicate the ordering of $s_i'$, where $V(s_1') < V(s_2') < V(s_3') \ldots < V(s_m')$. Then, the perturbed probability distribution is:

$$\widehat{\xi}_{\mathrm{Wang}}(s_i') \cdot \widehat{T}(s, a, s_i') = \begin{cases} g(\frac{1}{m}), & \text{if } i = 1. \\ g(\frac{i}{m}) - g(\frac{i-1}{m}), & \text{if } 1 < i < m. \\ 1 - g(\frac{i-1}{m}), & \text{if } i = m. \end{cases} \tag{12}$$

The transition distribution perturbed according to the Wang risk measure assigns gradually decreasing probability to higher-value successor states. The parameter $\eta$ controls the extent to which the probabilities are modified. For large $\eta >> 0$, almost all of the probability is placed on the worst outcome(s), and as $\eta \to 0$, the perturbed distribution approaches a uniform distribution.

# C Implementation Details

The code for the experiments can be found at github.com/marc-rigter/1R2R.

## C.1 Model Training

We represent the model as an ensemble of neural networks that output a Gaussian distribution over the next state and reward given the current state and action:

$$\widehat{T}_\phi(s', r | s, a) = \mathcal{N}(\mu_\phi(s, a), \Sigma_\phi(s, a)).$$

Following previous works [75, 76, 58], during model training we train an ensemble of 7 such dynamics models and pick the best 5 models based on the validation error on a held-out test set of 1000 transitions from the dataset $\mathcal{D}$. Each model in the ensemble is a 4-layer feedforward neural network with 200 hidden units per layer. $P(T \mid \mathcal{D})$ is assumed to be a uniform distribution over the 5 models. Increasing the number of networks in the ensemble has been shown to improve uncertainty estimation [44]. However, we used the same ensemble as previous works to facilitate a fair comparison.

The learning rate for the model training is 3e-4 for both the MLE pretraining, and it is trained using the Adam optimiser [36]. The hyperparameters used for model training are summarised in Table 4.

## C.2 Policy Training

We sample a batch of 256 transitions to train the policy and value function using soft actor-critic (SAC) [30]. We set the ratio of real data to $f = 0.5$ for all datasets, meaning that we sample 50% of the batch transitions from $\mathcal{D}$ and the remaining 50% from $\widehat{\mathcal{D}}$. We chose this value because we found it worked better than using 100% synthetic data (i.e. $f = 0$), and it was found to work well in previous works [75, 58]. We represent the $Q$-networks and the policy as three layer neural networks with 256 hidden units per layer.

For SAC [30] we use automatic entropy tuning, where the entropy target is set to the standard heuristic of $-\dim(A)$. The only hyperparameter that we modify from the standard implementation of SAC is the learning rate for the actor, which we set to 1e-4 as this was reported to work better in the CQL paper [40] which also utilised SAC. For reference, the hyperparameters used for SAC are included in Table 4.

## C.3 1R2R Implementation Details

When sampling successor states in Algorithm 1, we sample $m = 10$ candidate successor states to generate the set $\Psi$. For each run, we perform 2,000 iterations of 1,000 gradient updates. The rollout policy used to generate synthetic data is the current policy. The base hyperparameter configuration for 1R2R that is shared across all runs is shown in Table 4.

In our main experiments, the only parameters that we vary between datasets are the length of the synthetic rollouts ($k$), and the risk measure parameter. Details are included in the Hyperparameter Tuning section below. In Appendix D.4, we additionally include results where we increase $m$ from its base value, and increase the size of the ensemble.

## C.4 1R2R Hyperparameter Tuning

The hyperparameters that we tune for 1R2R are the length of the synthetic rollouts ($k$), and the parameter associated with the chosen risk measure ($\alpha$ for 1R2R$_{\text{CVaR}}$ or $\eta$ for 1R2R$_{\text{Wang}}$). For each dataset, we select the parameters that obtain the best online performance for the desired objective for that dataset. The objectives for each of the datasets are: optimising expected value for D4RL MuJoCo and optimising static CVaR$_{0.1}$ for all other domains.

The rollout length is tuned from $k \in \{1, 5\}$. For each of the two versions of our algorithm, 1R2R$_{\text{CVaR}}$ and 1R2R$_{\text{Wang}}$, the parameters are tuned from the following settings:

- 1R2R$_{\text{CVaR}}$: the rollout length, $k$, and CVaR parameter, $\alpha$, are selected from: $(k, \alpha) \in \{1, 5\} \times \{0.5, 0.7, 0.9\}$.

- 1R2R$_{\text{Wang}}$: the rollout length, $k$, and Wang risk parameter, $\eta$, are selected from: $(k, \eta) \in \{1, 5\} \times \{0.1, 0.5, 0.75\}$.

Note that while the risk measures are only moderately risk-averse, in the dynamic risk approach of 1R2R, they are applied recursively at each step. This can result in a high degree of risk-aversion over many steps [24].

Please refer to Table 5 for a list of the hyperparameters used for 1R2R for each dataset, and to Table 4 for the base set of hyperparameters that is shared across all datasets.

Table 4: Base hyperparameter configuration shared across all runs of 1R2R.

| | Hyperparameter | Value |
|---|---|---|
| SAC | critic learning rate | 3e-4 |
| | actor learning rate | 1e-4 |
| | discount factor ($\gamma$) | 0.99 |
| | soft update parameter ($\tau$) | 5e-3 |
| | target entropy | -dim($A$) |
| | batch size | 256 |
| 1R2R | no. of model networks | 7 |
| | no. of elites | 5 |
| | ratio of real data ($f$) | 0.5 |
| | no. of iterations ($N_{\text{iter}}$) | 2000 |
| | no. of candidate successor states ($m$) | 10 |
| | rollout policy | current $\pi$ |

Table 5: Hyperparameters used by 1R2R for each dataset, chosen according to best online performance. $k$ is the rollout length, $\alpha$ is the parameter for CVaR, and $\eta$ is the parameter for the Wang risk measure.

| | | 1R2R (CVaR) | | 1R2R (Wang) | |
|---|---|---|---|---|---|
| | | $k$ | $\alpha$ | $k$ | $\eta$ |
| Medium Random | HalfCheetah | 5 | 0.9 | 5 | 0.1 |
| | Hopper | 5 | 0.7 | 5 | 0.5 |
| | Walker2D | 5 | 0.9 | 5 | 0.1 |
| Medium Medium | HalfCheetah | 5 | 0.9 | 5 | 0.1 |
| | Hopper | 5 | 0.9 | 5 | 0.1 |
| | Walker2D | 5 | 0.9 | 5 | 0.1 |
| Medium Replay | HalfCheetah | 5 | 0.9 | 5 | 0.1 |
| | Hopper | 1 | 0.5 | 1 | 0.75 |
| | Walker2D | 1 | 0.9 | 1 | 0.1 |
| Medium Expert | HalfCheetah | 5 | 0.9 | 5 | 0.1 |
| | Hopper | 5 | 0.9 | 5 | 0.1 |
| | Walker2D | 1 | 0.7 | 1 | 0.5 |
| Medium | Hopper (Mod) | 5 | 0.9 | 5 | 0.1 |
| | Walker2D (Mod) | 1 | 0.9 | 1 | 0.5 |
| | Hopper (High) | 5 | 0.9 | 5 | 0.1 |
| | Walker2D (High) | 1 | 0.7 | 1 | 0.1 |
| Medium Replay | Hopper (Mod) | 5 | 0.7 | 5 | 0.5 |
| | Walker2D (Mod) | 1 | 0.9 | 1 | 0.1 |
| | Hopper (High) | 5 | 0.5 | 5 | 0.75 |
| | Walker2D (High) | 1 | 0.9 | 1 | 0.1 |
| Medium Expert | Hopper (Mod) | 5 | 0.7 | 5 | 0.75 |
| | Walker2D (Mod) | 1 | 0.5 | 1 | 0.75 |
| | Hopper (High) | 5 | 0.9 | 5 | 0.5 |
| | Walker2D (High) | 1 | 0.7 | 1 | 0.5 |
| | HIVTreatment | 1 | 0.7 | 1 | 0.5 |
| | CurrencyExchange | 1 | 0.5 | 1 | 0.75 |

# D  Additional Results

## D.1  MuJoCo Stochastic Expected Value Results

Table 6: Expected value results for the Stochastic MuJoCo benchmarks using the normalisation procedure proposed by [21]. We report the normalised performance during the last 10 iterations of training averaged over 5 seeds. $\pm$ captures the standard deviation over seeds. Highlighted numbers indicate results within 10% of the best score. "div." indicates that the value function diverged for at least one run.

| | | $1R2R_{CVaR}$ | $1R2R_{Wang}$ | Ablation | O-RAAC | CODAC | RAMBO | CQL | IQL | TD3+BC |
|---|---|---|---|---|---|---|---|---|---|---|
| Medium | Hopper (Mod) | $67.2 \pm 5.7$ | $65.9 \pm 7.0$ | $64.1 \pm 10.0$ | $15.6 \pm 9.4$ | $50.7 \pm 3.0$ | $72.9 \pm 14.7$ | $50.2 \pm 0.6$ | $50.6 \pm 4.1$ | $48.4 \pm 1.4$ |
| | Walker2D (Mod) | $67.3 \pm 7.8$ | $62.2 \pm 6.3$ | $27.6 \pm 23.4$ | $38.8 \pm 7.1$ | $44.1 \pm 1.5$ | $34.5 \pm 2.2$ | $58.4 \pm 1.6$ | $45.6 \pm 1.3$ | $41.4 \pm 17.4$ |
| | Hopper (High) | $57.1 \pm 4.5$ | $58.1 \pm 5.9$ | $53.0 \pm 4.7$ | $28.8 \pm 11.0$ | $69.9 \pm 2.1$ | $54.4 \pm 4.4$ | $74.7 \pm 5.1$ | $61.6 \pm 8.0$ | $66.3 \pm 1.3$ |
| | Walker2D (High) | $59.5 \pm 11.8$ | $58.8 \pm 8.9$ | div. | $41.6 \pm 9.6$ | $65.7 \pm 0.6$ | $56.9 \pm 3.5$ | $75.2 \pm 1.3$ | $54.8 \pm 2.0$ | $77.6 \pm 2.3$ |
| Medium Replay | Hopper (Mod) | $83.6 \pm 18.6$ | $76.8 \pm 9.9$ | $64.5 \pm 9.8$ | $15.8 \pm 3.0$ | $61.2 \pm 10.7$ | $67.6 \pm 11.3$ | $66.9 \pm 10.8$ | $-0.5 \pm 0.0$ | $38.2 \pm 4.9$ |
| | Walker2D (Mod) | $53.9 \pm 9.4$ | $51.7 \pm 11.4$ | $57.4 \pm 3.0$ | $24.2 \pm 6.8$ | $47.8 \pm 19.7$ | $63.6 \pm 4.5$ | $50.1 \pm 1.9$ | $16.2 \pm 2.8$ | $48.4 \pm 1.5$ |
| | Hopper (High) | $114.3 \pm 10.8$ | $102.0 \pm 9.1$ | $48.4 \pm 16.0$ | $26.2 \pm 16.0$ | $54.5 \pm 2.9$ | $54.1 \pm 19.7$ | $96.7 \pm 13.0$ | $15.1 \pm 9.6$ | $58.4 \pm 6.1$ |
| | Walker2D (High) | $84.1 \pm 5.3$ | $76.4 \pm 6.0$ | $75.4 \pm 8.0$ | $19.1 \pm 4.7$ | $80.4 \pm 11.6$ | $84.1 \pm 4.0$ | $61.1 \pm 3.0$ | $-0.1 \pm 0.8$ | $73.4 \pm 2.3$ |
| Medium Expert | Hopper (Mod) | $94.1 \pm 17.2$ | $71.3 \pm 42.2$ | $64.0 \pm 41.7$ | $32.8 \pm 8.7$ | $96.6 \pm 8.8$ | $71.9 \pm 20.7$ | $106.7 \pm 3.8$ | $107.5 \pm 7.7$ | $100.9 \pm 2.0$ |
| | Walker2D (Mod) | $72.2 \pm 7.3$ | $77.3 \pm 9.1$ | div. | $27.6 \pm 12.4$ | $81.7 \pm 4.8$ | $58.0 \pm 22.3$ | $94.2 \pm 1.1$ | $63.2 \pm 1.7$ | $93.5 \pm 0.9$ |
| | Hopper (High) | $54.1 \pm 5.5$ | $63.2 \pm 11.0$ | $48.4 \pm 7.4$ | $18.9 \pm 11.2$ | $79.9 \pm 3.5$ | $60.0 \pm 6.1$ | $88.3 \pm 3.5$ | $64.0 \pm 16.5$ | $89.9 \pm 1.4$ |
| | Walker2D (High) | $49.9 \pm 15.4$ | $44.4 \pm 17.6$ | div. | $27.3 \pm 13.4$ | $79.5 \pm 18.0$ | $31.1 \pm 14.9$ | $104.3 \pm 2.3$ | $72.7 \pm 4.9$ | $107.0 \pm 2.3$ |

## D.2  HIV Treatment Return Distributions

Figure 3 illustrates the return distributions for each algorithm from all evaluation episodes aggregated over 5 seeds per algorithm. Note that in the ablation in Figure 3c, training became unstable for a subset of the runs. This resulted in very poor performance for a subset of the runs. Because the plots in Figure 3 are aggregated across all 5 runs for each algorithm, this results in the strongly multi-modal performance observed for the ablation in Figure 3c.

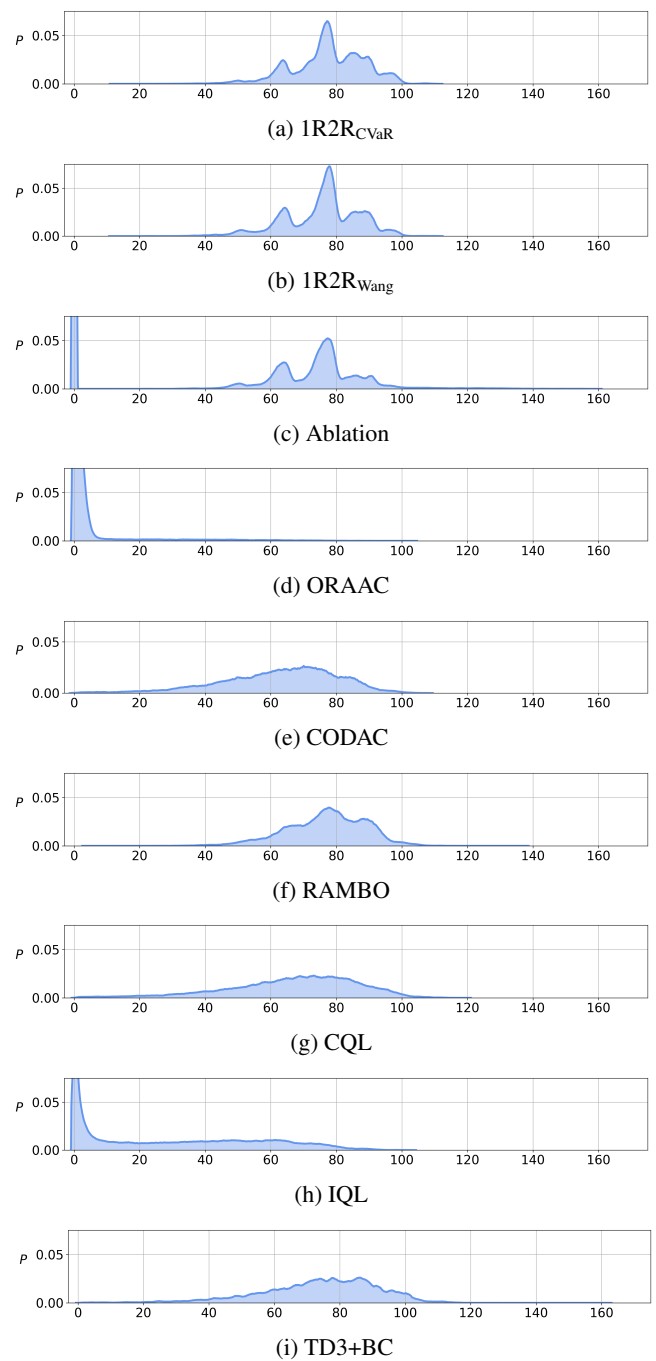

Figure 3: Return distributions for each algorithm on the HIV Treatment domain. The plots are generated by aggregating all evaluation episodes. There are 200 evaluation episodes per iteration for each of the last 10 iterations of training, across 5 seeds per algorithm.

## D.3 Currency Exchange Return Distributions

To compare risk-averse behaviour to risk-neutral behaviour on the Currency Exchange domain, we plot the return distributions for $1R2R_{CVaR}$ and RAMBO in Figure 4. We observe that RAMBO performs well in expectation, but has a long tail of poor returns, resulting in poor performance for static $CVaR_{0.1}$. 1R2R performs less well in expectation, but there is less variability in the returns, resulting in better performance for static $CVaR_{0.1}$.

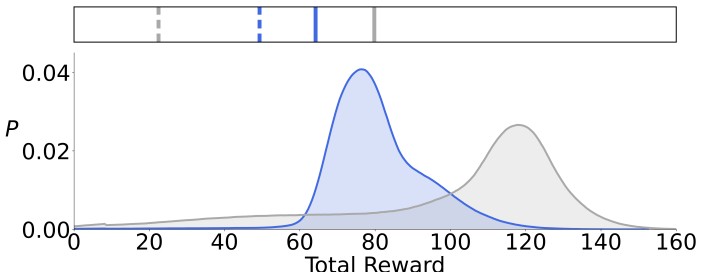

Figure 4: Return distributions for the Currency Exchange domain, aggregated over 5 seeds. Blue: $1R2R_{CVaR}$, Grey: RAMBO. Solid vertical bars indicate average return, dashed vertical bars indicate the mean of the worst 10% of returns (i.e. static $CVaR_{0.1}$). $1R2R_{CVaR}$ performs better for static $CVaR_{0.1}$ while RAMBO performs better in expectation.

## D.4 Results with Modified Parameters

In Table 7 we present additional results for 1R2R using parameter values that have been modified from those used in our main experiments. Due to computational limitations, we focus on $1R2R_{CVaR}$ on the Stochastic Walker2D domains.

The results in Table 7 show that by either increasing the number of networks in the model ensemble ($1R2R_{CVaR}$ (Larger Ensemble)), or increasing the number of samples of successor states drawn from the model ($1R2R_{CVaR}$ (More Samples)), similar or slightly worse performance is obtained compared to the base version of our algorithm. This indicates that it is not possible to improve the performance of 1R2R by naively increasing the computation available.

In the base configuration of our algorithm we set the ratio of real data sampled during policy training to $f = 0.5$, as described in Appendix C.2. The results in Table 7 show that the performance degrades significantly when training the policy and value function entirely from synthetic samples generated by the model ($1R2R_{CVaR}$ (No Real Data)). This indicates that sampling training data from the real dataset in addition to synthetic data is crucial to obtaining strong performance. This is probably because sampling data from the real dataset helps to mitigate errors due to inaccuracies in the model. As discussed in Section 6, the model may be somewhat inaccurate for these domains due to the restriction of using Gaussian transition functions.

Table 7: Additional results using different parameter values for $1R2R_{CVaR}$. The table includes results for both the Expected Value and the $CVaR_{0.1}$ objective, as indicated by the Objective column. The column $1R2R_{CVaR}$ (Base) shows the results for $1R2R_{CVaR}$ using the base algorithm configuration described in Appendix C. In the remaining columns, the algorithm is modified as follows. $1R2R_{CVaR}$ (Larger Ensemble) uses an ensemble of 20 total neural networks with the best 15 selected. $1R2R_{CVaR}$ (No Real Data) does not sample any real data when performing updates (i.e. the ratio of real data is $f = 0$). $1R2R_{CVaR}$ (More Samples) samples 50 successor states when generating the set of successor states in Line 8 of Algorithm 1 (i.e. $m = 50$).

| | | Objective | $1R2R_{CVaR}$ (Base) | $1R2R_{CVaR}$ (Larger Ensemble) | $1R2R_{CVaR}$ (No Real Data) | $1R2R_{CVaR}$ (More Samples) |
|---|---|---|---|---|---|---|
| Medium | Walker2D (Mod) | $CVaR_{0.1}$ | $29.9 \pm 8.7$ | $27.5 \pm 7.5$ | $15.0 \pm 7.7$ | $26.6 \pm 5.1$ |
| | | Expected Value | $67.3 \pm 7.8$ | $60.7 \pm 9.9$ | $34.0 \pm 13.3$ | $58.2 \pm 6.9$ |
| | Walker2D (High) | $CVaR_{0.1}$ | $19.4 \pm 12.6$ | $4.1 \pm 4.7$ | $0.6 \pm 0.7$ | $9.5 \pm 1.0$ |
| | | Expected Value | $59.5 \pm 11.8$ | $38.7 \pm 24.4$ | $28.0 \pm 4.9$ | $51.4 \pm 9.7$ |
| Medium Replay | Walker2D (Mod) | $CVaR_{0.1}$ | $30.1 \pm 7.2$ | $27.9 \pm 3.5$ | $10.8 \pm 14.4$ | $26.8 \pm 6.0$ |
| | | Expected Value | $53.9 \pm 9.4$ | $53.3 \pm 7.2$ | $30.9 \pm 19.7$ | $51.2 \pm 8.4$ |
| | Walker2D (High) | $CVaR_{0.1}$ | $46.0 \pm 10.1$ | $39.1 \pm 6.9$ | $1.8 \pm 1.7$ | $39.6 \pm 6.5$ |
| | | Expected Value | $84.1 \pm 5.3$ | $81.8 \pm 4.8$ | $26.4 \pm 10.8$ | $79.7 \pm 7.0$ |
| Medium Expert | Walker2D (Mod) | $CVaR_{0.1}$ | $34.8 \pm 6.5$ | $31.7 \pm 6.5$ | $7.7 \pm 5.6$ | $42.9 \pm 16.7$ |
| | | Expected Value | $72.2 \pm 7.3$ | $73.5 \pm 7.2$ | $22.9 \pm 9.6$ | $79.3 \pm 11.0$ |
| | Walker2D (High) | $CVaR_{0.1}$ | $10.8 \pm 4.3$ | $11.5 \pm 1.7$ | $7.3 \pm 5.4$ | $12.3 \pm 0.6$ |
| | | Expected Value | $49.9 \pm 15.4$ | $52.7 \pm 9.9$ | $21.3 \pm 11.2$ | $50.4 \pm 10.5$ |

# E    Experiment Details

## E.1    Computation Requirements

Each run of 1R2R requires 16-24 hours of computation time on a system with an Intel Core i7-8700 CPU @ 3.20GHz CPU and an NVIDIA GeForce GTX 1070 graphics card.

## E.2    Domains

The D4RL MuJoCo domains are from the D4RL offline RL benchmark [21]. In this work, we introduce the following domains.

**Currency Exchange**    This domain is an adaptation of the Optimal Liquidation Problem [2, 8] to offline RL. We assume that the agent initially holds 100 units of currency A, and wishes to convert all of this to currency B before a deadline $T$, subject to an exchange rate which changes stochastically. At each time step, the agent may choose how much (if any) of currency A to convert to currency B at the current exchange rate.

An aggressive strategy is to delay converting the currency, in the hope that random fluctuations will lead to a more favourable exchange rate before the deadline $T$. However, this may lead to a poor outcome if the exchange rate decreases and fails to recover before the deadline. More risk-averse strategies are to either: a) gradually convert the currency over many steps, or b) immediately convert all of the currency even if the current exchange rate is mediocre, to avoid the risk of having to convert the money at even worse exchange rate in the future, to meet the deadline $T$.

The state space is 3-dimensional, consisting of: the timestep, $t \in \{0, 1, 2, \ldots, T\}$; the amount of currency A remaining, $m \in [0, 100]$; and the current exchange rate, $p \in [0, \infty)$. We assume that the exchange rate evolves according to an Ornstein-Uhlenbeck process, which is commonly used for financial modelling [49]:

$$dp_t = \theta(\mu - p_t)dt + \sigma dW_t, \tag{13}$$

where $W_t$ denotes the Wiener process. In our implementation we set: $\mu = 1.5$, $\sigma = 0.2$, $\theta = 0.05$. The exchange rate at the initial time step is $p_0 \sim \mathcal{N}(1, 0.05^2)$. The action space is single dimensional, $a \in [-1, 1]$. We assume that a positive action corresponds to the proportion of $m$ to convert to currency B at the current step, while a negative action corresponds to converting none of the currency. The reward is equal to the amount of currency B received at each step.

To generate the dataset, we generate data from a random policy that at each step chooses a uniform random proportion of currency to convert with probability 0.2, and converts none of the currency with probability 0.8.

**HIV Treatment**    The environment is based on the implementation in [26] of the physical model described by [20]. The patient state is a 6-dimensional continuous vector representing the concentrations of different types of cells and viruses in the patients blood. There are two actions, corresponding to giving the patient Reverse Transcriptase Inhibitors (RTI) or Protease Inhibitors (PI) (or both). Unlike [20] which considered discrete actions, we modify the domain to allow for continuous actions representing variable quantities of each drug. Following previous work [33], we assume that the

Table 8: Expected return received by a randomly initialised policy and an expert policy trained online using SAC. The expert SAC policy is trained online for 2e6 updates for the Stochastic MuJoCo domains, and 5e6 updates for HIVTreatment. In Tables 1, 2, and 3 the values reported are normalised between 0 and 100 using the procedure from [21] and the values in this table.

|                            | Random Policy | Expert Policy |
|----------------------------|---------------|---------------|
| Hopper (Moderate-Noise)    | 20.1          | 3049.4        |
| Walker2D (Moderate-Noise)  | -1.9          | 4479.2        |
| Hopper (High-Noise)        | 11.2          | 2035.1        |
| Walker2D (High-Noise)      | -5.3          | 4101.3        |
| HIVTreatment               | -48.7         | 5557.9        |
| CurrencyExchange           | 0.0           | 135.0         |

treatment at each time step is applied for 20 days, and that there are 50 time steps. Therefore, each episode corresponds to a total period of 1000 days. Also following [33], we introduce stochasticity into the domain by adding noise to the efficacy of each drug ($\epsilon_1$ and $\epsilon_2$ in [20]). The noise added to the efficacy is normally distributed, with a standard deviation of 15% of the standard value.

For this domain, the dataset is constructed in the same manner as the *Medium-Replay* datasets from D4RL [21]. Specifically, a policy is trained with SAC [30] until it reaches $\approx 40\%$ of the performance of an expert policy. The replay buffer of data collected until reaching this point is used as the offline dataset.

The reward function is that used by [26], except that we scale the reward by dividing it by 1e6. A low concentration of HIV viruses is rewarded, and a cost is associated with the magnitude of treatment applied, representing negative side effects. The average reward received by a random policy and an expert policy trained online using SAC [30] can be found in Table 8. In the paper, values are normalised between 0 and 100 using the method proposed by [21] and the values in Table 8. Thus, a score of 100 represents the average reward of an expert policy, and 0 represents the average reward of a random policy.

**Stochastic MuJoCo**   These benchmarks are obtained by applying random perturbation forces to the original *Hopper* and *Walker2D* MuJoCo domains. We chose to focus on these two domains as there is a risk of the robot falling over before the end of the episode. The perturbation forces represent disturbances due to strong wind or other sources of uncertainty. The force is applied in the $x$-direction at each step, according to a random walk on the interval $[-f_{\mathrm{MAX}}, f_{\mathrm{MAX}}]$. At each step, the change in force relative to the previous step is sampled uniformly from $\Delta f \sim Unif(-0.1 \cdot f_{\mathrm{MAX}}, 0.1 \cdot f_{\mathrm{MAX}})$. This is motivated by the fact that the strength of wind is often modelled as a random walk [52].

We test using two levels of noise applied. For the *Moderate-Noise* datasets, the level of noise results in a moderate degradation in the performance of an online SAC agent, compared to the original deterministic domain. For the *High-Noise* datasets, the level of noise results in a significant degradation in the performance of an online SAC agent. The value of $f_{\mathrm{MAX}}$ is set to the following in each of the domains:

- *Hopper Moderate-Noise*: $f_{\mathrm{MAX}} = 2.5$ Newtons.
- *Hopper High-Noise*: $f_{\mathrm{MAX}} = 5$ Newtons.
- *Walker2D Moderate-Noise*: $f_{\mathrm{MAX}} = 7$ Newtons.
- *Walker2D High-Noise*: $f_{\mathrm{MAX}} = 12$ Newtons.

We construct three datasets: *Medium*, *Medium-Replay*, and *Medium-Expert*. These are generated in the same manner as the D4RL datasets [21]. The expert policy is a policy trained online using SAC until convergence. The medium policy is trained until it obtains $\approx 40\%$ of the performance of the expert policy. The *Medium-Replay* dataset consists of the replay buffer collected while training the medium policy. The *Medium* dataset contains transitions collected from rolling out the medium policy. The *Medium-Expert* dataset contains a mixture of transitions generated by rolling out both the medium and the expert policy.

Following previous works [68, 47], we assume that the objective on all of the stochastic domains sis to optimise the static $\mathrm{CVaR}_{0.1}$ (i.e. the average total reward in the worst 10% of runs).

### E.3   Evaluation Procedure

For D4RL MuJoCo, we evaluate the policy for 10 episodes at each of the last 10 iterations of training, for each of 5 seeds. The average normalised total reward received throughout this evaluation is reported in the D4RL results table (Table 1).

For Stochastic MuJoCo, we evaluate the policy for 20 episodes at each of the last 10 iterations of training. $\mathrm{CVaR}_{0.1}$ is computed by averaging the total reward in the worst two episodes at each iteration. The value of $\mathrm{CVaR}_{0.1}$ is then averaged across the last 10 iterations, and across each of the 5 seeds. These are the results reported for $\mathrm{CVaR}_{0.1}$ in Table 3. For HIVTreatment and CurrencyExchange, we evaluate the policy for 200 episodes at each of the last 10 iterations of training, for each of 5 seeds. The average total reward for the worst 20 episodes at each evaluation is used to compute $\mathrm{CVaR}_{0.1}$.

## E.4  Baselines

We compare 1R2R against offline RL algorithms designed for risk-sensitivity (ORAAC [68] and CODAC [47]) in addition to performant risk-neutral model-based (RAMBO [58], COMBO [75], and MOPO [76]) and model-free (IQL [38], TD3+BC [22], CQL [40], and ATAC [11]) algorithms.

For the baselines, we use the following implementations:

- O-RAAC: github.com/nuria95/O-RAAC
- CODAC: github.com/JasonMa2016/CODAC
- RAMBO: github.com/marc-rigter/rambo
- IQL: github.com/ikostrikov/implicit_q_learning
- TD3+BC: github.com/sfujim/TD3_BC
- CQL and COMBO: github.com/takuseno/d3rlpy
- ATAC: https://github.com/microsoft/lightATAC
- MOPO: https://github.com/tianheyu927/mopo

The results for D4RL MuJoCo-v2 for RAMBO, IQL, ATAC, COMBO, MOPO and TD3+BC are taken from the original papers. Because the original CQL paper used the -v0 domains, we report the results for CQL on the -v2 domains from [69]. The IQL papers does not report results for the *random* datasets, so we generate the results for IQL in the random datasets by running the official implementation algorithm ourselves. The results for CODAC in the original paper use D4RL MuJoCo-v0, and O-RAAC does not report results on the D4RL benchmarks. Therefore, we obtain the results for D4RL MuJoCo-v2 for CODAC and O-RAAC by running the algorithms ourselves.

For the stochastic domains (Currency Exchange, HIV Treatment, and MuJoCo Stochastic) all of the results were obtained by running the algorithms ourselves.

## E.5  Baseline Hyperparameter Tuning

We tune each of the algorithms online to achieve the best performance for the desired objective in each dataset. The desired optimisation objectives in each domain are: expected value in D4RL MuJoCo and static $\text{CVaR}_{0.1}$ in all other domains. O-RAAC and CODAC can be configured to optimise a static risk measure, or expected value. Therefore, we set O-RAAC and CODAC to optimise the appropriate objective for each dataset.

For each of the baseline algorithms, we tune the following parameters based on the best online performance (averaged over 5 seeds) for the target objective. Where a hyperparameter value is not specified, it was set to the default value in the corresponding original paper.

- CODAC: The risk measure to optimise is set to the desired objective (Expected value/$\text{CVaR}_{0.1}$). For Stochastic MuJoCo and HIVTreatment, we tune the parameters in the same manner is the original paper [47], choosing $(\omega, \zeta) \in \{0.1, 1, 10\} \times \{-1, 10\}$. For Stochastic MuJoCo and HIVTreatment we use the default critic learning rate of $\eta_{\text{critic}} = 3e - 5$. To generate the results for CODAC on D4RL MuJoCo-v2, we use the same parameters as reported in the paper for D4RL MuJoCo-v0.

- O-RAAC: The risk measure to optimise is set to the desired objective (Expected value/$\text{CVaR}_{0.1}$). Following the original paper [68] we tune the imitation learning weighting: $\lambda \in \{0.25, 0.5\}$.

- RAMBO: We tuned the rollout length and adversarial weighting from: : $(k, \lambda) \in \{(1, 3e - 4), (2, 3e - 4), (5, 3e - 4), (5, 0)\}$. This is the same procedure as in the original paper [58], except that we additionally included a rollout length of 1 as we found this was necessary for RAMBO to train stably on some of the stochastic domains.

- IQL: For Stochastic MuJoCo, we used the same parameters as for D4RL MuJoCo in the original paper [38]. For HIVTreatment and Currency Exchange, we chose the best performance between the parameters for D4RL MuJoCo and D4RL AntMaze from the original paper.

- CQL: Following the original paper [40], we tuned the Lagrange parameter, $\tau \in \{5, 10\}$. All other parameters are set to their default values.

- TD3+BC: Following the original paper, we set $\alpha$ to 2.5 [22].

- MOPO: The rollout length and conservative weighting are each tuned within $\{1, 5\}$ following the original paper [76].

- COMBO: The conservative weighting was tuned within $\{0.5, 1, 5\}$ following the original paper [75].

- ATAC: $\beta$ was tuned within $\{0, 4^{-4}, 4^{-2}, 1, 4^2, 4^4\}$.

