# OpenReview forum: "One Risk to Rule Them All: A Risk-Sensitive Perspective on Model-Based Offline Reinforcement Learning"
_NeurIPS.cc/2023/Conference — NeurIPS 2023 poster_

### Official Review · Reviewer_unPm · 2023-07-04

**Soundness:** 2 fair
**Presentation:** 3 good
**Contribution:** 3 good
**Rating:** 6
**Confidence:** 4

**Summary:**

This paper proposes a new model-based offline RL algorithm that learns a policy which is risk-averse (wrt aleatoric uncertainty measured by some dynamic risk measure) and pessimistic (wrt epistemic uncertainty from modeling). First, the algorithm learns a posterior distribution over MDP transitions given the dataset, using an ensemble of neural nets. Then, the algorithm samples successor states from the worst-case perturbation of the learned model, effectively modeling the Bellman equation for dynamic risk.

**Strengths:**

The method is clean and the high-level ideas are clearly explained. Extensive experiments are also encouraging and paper is pretty well written.

**Weaknesses:**

1. My understanding is that this method is targeting the dynamic risk of the coherent risk measure with envelope B_p. However, all of the evaluations, as well as most of the paper, seem to suggest that 1R2R is a good algorithm for risk-neutral and static risk objectives (which the authors indeed show in experiments). It's not clear to me why optimizing for the dynamic risk should result in good performance for risk-neutral or static risk objectives, and so the method seems more hand-wavy. For example, if we assume that model learning succeeds, can we prove any PAC bounds wrt risk-neutral or static risk objectives?
2. It would be interesting to have some ablations on the success of 1R2R: are the improvements in performance mostly due to pessimism wrt aleatoric uncertainty, or epistemic uncertainty, or simply model-based offline RL?

Also, please see Questions section.

**Questions:**

1. Why is Line 184 labeled "Problem 1"? I don't see a Problem 2.
2. Why do you take on the Bayesian perspective for learning \bar T? Another way of learning the model is the MLE, ie train a single neural net to maximize log likelihood of successor state, so I'm wondering why you choose to learn P(T|D) and then derive \bar T from that?
3. How is the adversarial perturbation actually computed in practice (Line 10)? Does you need to perform a two-stage optimization procedure?
4. How are the hyperparameters of the method selected, and how many online samples were used for hyperparameter tuning? (In theory, papers about offline RL should only be using offline samples for algorithm design, but in practice, this is almost always violated. So, it would be nice to report how many online samples were used, especially given that this paper introduces many more hyperparameters (ie Table 4)).
5. Have you compared with ATAC (Adversarially Trained Actor Critic for Offline Reinforcement Learning), which is one of the SOTA offline RL methods?

**Limitations:**

Please see weaknesses/questions.

---

> ### Author Rebuttal · Authors · 2023-08-09
>
> Thank you for the valuable time you have spent reviewing our paper. We respond to your comments and questions below.
>
> ### Theory and Dynamic vs Static Risk
>
> We have **added theoretical justification** for why our approach avoids both epistemic and aleatoric uncertainty in Proposition 1 in the response to Reviewer RPWe.
>
> As you point out, our approach utilises dynamic risk and we evaluate our approach using static risk objectives. This is necessary because we cannot evaluate the performance for dynamic risk in practice. In Lines 355-365 of the paper we discuss this limitation.
>
> Dynamic and static risk can be related in the following way. Dynamic risk is equivalent to adversarially modifying the transition dynamics *independently* at every time step according to the risk envelope. Static risk is equivalent to adversarially modifying the transition dynamics at each time step subject to the constraint that the perturbation to each *entire trajectory* remains within the risk envelope (see e.g. [1, 2]). Thus, the optimal behaviour with respect to both static and dynamic risk is the optimal behaviour in an adversarially perturbed MDP, where the worst transitions are made more likely. This is why we expect optimal behaviour for dynamic risk to be similar to that for static risk.
>
> We are unaware of PAC bounds relating the two. We think this is an exciting direction for future work.
>
> [1] Chow, Yinlam, et al. "Risk-sensitive and robust decision-making: a CVaR optimization approach." NeurIPS, 2015.
>
> [2] Rigter, Marc, et al. "Planning for risk-aversion and expected value in MDPs." ICAPS, 2022.
>
> ### Additional Ablations
>
> As you have suggested, we have **expanded the ablations** in Tables 1 and 2 of the global response.
>
> The two key components of our approach are 1) risk-averse sampling and 2) using an ensemble of models. In the new results, we separately ablate each of these components.
>
> When either of these components is removed, we observe that training may diverge. This illustrates that both components are necessary to prevent value function instability resulting from backing-up out-of-distribution value estimates (i.e. avoiding epistemic uncertainty). The ablations also show that risk-averse sampling is necessary to achieve strong risk-averse performance to aleatoric uncertainty on the stochastic Currency Exchange domain.
>
> We have also added comparisons to more model-based RL algorithms, COMBO and MOPO. We observe that 1R2R easily outperforms these algorithms on the stochastic domains (Currency Exchange and HIV Treatment). This indicates that the strong performance of 1R2R is not solely due to being model-based.
>
> ### Question 1.
>
> In the final version we will use the name “Dynamic Risk Optimisation in Bayesian MDP” for our problem formulation, to make this clearer.
>
> ### Question 2.
>
> We define $\overline{T}$ using a distribution over models, $P(T | D)$, so that we can relate learning in an ensemble of models to risk-sensitivity. Recall that risk-measures require a *distribution* as input (they map distributions to scalar values), and that we require an ensemble of models to capture epistemic uncertainty. Given our ensemble of models, we need to be able to define $\overline{T}$: the distribution over successor states given the model ensemble. We do this using Equation 5 in the paper, which defines $\overline{T}$ using $P(T | D)$.
>
> In our practical implementation (Lines 246-247), we use a finite set of $M$ neural networks to define the model ensemble, and each of these networks is separately trained using MLE. We assume that $P(T | D)$ is a uniform distribution over these $M$ networks, i.e. $P(T | D) = \frac{1}{M}$ for all $T$ in the ensemble. This assumption means that when sampling from $\overline{T}$ in Line 8 of Algorithm 1, we first uniformly sample from the $M$ models in the ensemble, and then sample a successor from that model.
>
> ### Question 3.
>
> The optimisation problem in Line 10 of Algorithm 1 is over a discrete set of samples, making it straightforward. We first sort the samples from lowest to highest value successor states. Then, for the risk measures we consider (CVaR and Wang) the solution is computed in closed-form using Equations 11 and 12 in Appendix A.1.
>
> ### Question 4.
>
> To choose the hyperparameters, we evaluate each policy for 10 episodes in D4RL and 20 episodes in the stochastic domains at the end of training (for 5 seeds). Then, we choose the hyperparameters that obtained the best performance for the desired objective. We also optimise the hyperparameters for the baselines in this manner.
>
> ### Question 5. Additional Baselines
>
> In the Global Response, we have added **additional results** for ATAC as well as MOPO and COMBO (as requested by Reviewer RPWe). We observe that MOPO performs very poorly on all domains. 1R2R only slightly outperforms ATAC and COMBO on the D4RL domains. However, on the stochastic domains (Currency Exchange and HIV Treatment) 1R2R easily outperforms both ATAC and COMBO.
>
> This demonstrates that 1R2R outperforms current state-of-the-art algorithms. It also highlights that we should not rely on benchmarking only on deterministic environments.
>
> **Experiment Details**
>
> The D4RL results for ATAC, MOPO, and COMBO are from the original papers.
>
> For Currency Exchange and HIV Treatment, we ran the algorithms ourselves. For ATAC and MOPO we used the official implementations. For COMBO, there is no official implementation so we used the implementation from d3rlpy.
>
> We tuned the hyperparameters to obtain the best performance for the CVaR objective:
>
> - MOPO: rollout length and conservative weighting each tuned within {1, 5} following the original paper.
> - COMBO: the conservative weighting was tuned within {0.5, 1, 5} following the original paper.
> - ATAC: $\beta$ was tuned within {$0, 4^{-4}, 4^{-2}, 1, 4^2, 4^4$}.

---

### Official Review · Reviewer_8yT3 · 2023-07-06

**Soundness:** 3 good
**Presentation:** 3 good
**Contribution:** 3 good
**Rating:** 6
**Confidence:** 4

**Summary:**

> Our concerns are addressed by the authors. Thanks for your effort. We will upgrade the rating to weak accept.

This paper introduces a model-based risk-averse algorithm that utilizes risk aversion as a mechanism to jointly address the distributional shift problem and risk-related decision-making problems in risk-sensitive offline RL. The authors employ a risk-averse risk measure to simultaneously perturb the belief distribution and transition functions in an adversarial manner, enabling risk aversion towards both epistemic uncertainty and aleatoric uncertainty. Risk aversion to epistemic uncertainty reduces transition probabilities to state-action pairs that are out-of-distribution, and risk aversion to aleatoric uncertainty helps avoid actions that are inherently risky. The authors conduct experiments in both deterministic and stochastic environments, confirming the superior performance of the 1R2R algorithm.

**Strengths:**

1. The paper is well written. The original contributions are highlighted clearly.
2. The paper provides a concise and understandable introduction to the background and related work. It is highly reader-friendly, ensuring ease of comprehension for the intended readers.
3. The paper demonstrates a well-organized structure, and the approach of utilizing risk aversion to address the distributional shift problem is innovative and holds theoretical viability.


**Weaknesses:**

1. While adopting a joint risk-aversion mechanism for both epistemic uncertainty and aleatoric uncertainty is simple and efficient, I think this approach is less flexible compared to previous methods which can address distributional shift and risk-sensitivity separately.

2. The design of ablation experiment merely removes the risk-sensitivity, which I think to be overly simplistic. Ablation Experiments part requires appropriate expansion, such as comparing the computational costs with the approaches using distributional value functions, and comparing the performance between using dynamic risk measures and using static risk measures.

3. I think 1R2R does not have a significant advantage over other baselines (i.e., RAMBO). Can the authors discuss more in-depth regarding this issue?

**Questions:**

1. How does the argmin is performed in Line 10 of Algorithm 1?
2. In my opinion, instead of characterizing this approach as risk-aversion methods, I think it resembles a conservative value update method. By introducing adversarial perturbations to the transition function, it increases the likelihood of transitioning to low-value successor states. Can the author clarify this?
3. For the experimental results of D4RL MuJoCo, how can you validate that the performance improvement of the 1R2R algorithm is indeed due to addressing the 'distributional shift' problem?
4. In Line 139, the equation is written as ($Z_{\mathrm{MDP}}=\sum_{t=0}^{\infty} \gamma R\left(s_t, a_t\right)$). Shouldn’t this be ($Z_{\mathrm{MDP}}=\sum_{t=0}^{\infty} \gamma^{t} R\left(s_t, a_t\right)$)?

**Limitations:**

This approach is less flexible compared to the previous methods which can address distributional shift and risk-sensitivity separately.

---

> ### Author Rebuttal · Authors · 2023-08-09
>
> Thank you for the valuable time you have spent reviewing our paper. We respond to your comments and questions below.
>
> ### Flexibility of the Approach
>
> We agree that our approach is simpler yet less flexible than existing approaches that can separately adjust how they address each source of uncertainty. In Lines 370-374, we discuss this limitation. We also discuss that a straightforward extension of our work is to apply *composite* risk measures [1]. Composite risk measures allow the aversion to aleatoric vs epistemic uncertainty to be adjusted separately. This is a direction we wish to explore in future work.
>
> [1] Eriksson, Hannes, et al. "Sentinel: taming uncertainty with ensemble based distributional reinforcement learning." *UAI*, 2022.
>
> ### Ablations
>
> As you have suggested, we have **expanded the ablations** in Tables 1 and 2 of the global response.
>
> The two key components of our approach are 1) risk-averse sampling and 2) using an ensemble of models. We include an additional ablation so that we separately ablate each of these components.
>
> When either of these components is removed, we observe that training may diverge. This illustrates that both components of our approach are necessary to prevent value function instability resulting from backing-up out-of-distribution value estimates (i.e. avoiding epistemic uncertainty). The ablations also show that risk-averse sampling is necessary to achieve strong risk-sensitive performance to aleatoric uncertainty on the stochastic Currency Exchange domain.
>
> ### Advantage Over Baselines
>
> On the D4RL domains (which are deterministic) our approach obtains similar to performance to some of the strongest baselines such as RAMBO. However, in the stochastic Currency Exchange domain (where the objective is to optimise CVaR) only 1R2R and CODAC are able to achieve strong risk-averse performance. In this domain, RAMBO performs well for average performance but not for the desired objective of CVaR. CODAC performs poorly on most of the other benchmarks. Thus, 1R2R is the only algorithm that achieves strong performance in the deterministic domains, as well as generating performant risk-averse behaviour in the stochastic domains.
>
> ### Question 1.
>
> Because the minimisation is performed over a discrete set of candidate successor states, it is a straightforward optimisation problem. To perform the minimisation, we first sort the sampled successor states from lowest to highest value. Then, the solution is computed in closed-form according to Equations 11 and 12 in Appendix A.1 for CVaR and the Wang risk measure, respectively.
>
> ### Question 2.
>
> Conservative value function methods (e.g. [2]) aim to learn a value function that lower-bounds the value function in the true environment. Risk-measures consider mappings from *distributions* to scalar values. This means that risk-measures can be viewed as the expectation under an adversarially perturbed distribution [3]. Our approach optimises the policy under an adversarially perturbed transition function, which is why we characterise it as a risk-sensitive approach.
>
> Under some conditions, it may also be the case that the value function that our algorithm computes is a lower bound on the true value function. If this is the case, our approach can also be considered conservative in this sense. Exploring this connection is an exciting direction for future work.
>
> [2] Kumar, Aviral, et al. "Conservative Q-learning for offline reinforcement learning." NeurIPS, 2020.
>
> [3] Artzner, Philippe, et al. "Coherent measures of risk." Mathematical finance, 1999.
>
> ### Question 3.
>
> If we naively apply online RL algorithms in the offline setting, the RL algorithm will back-up values estimated for out-of-distribution (OOD) state-action pairs, where the value function is inaccurate. This is referred to as the “distributional shift” problem. Because RL algorithms choose the highest value actions, this leads to the policy selecting OOD state-action pairs where the Q-value is overestimated. Repeatedly performing Bellman backups in this manner often leads to the value function diverging (see [4], Figure 1).
>
> In the results for our algorithm, we do not observe this issue of the value function diverging. However, if we ablate either component of our approach (either the ensemble or risk-averse sampling), then for some datasets the value function is unstable and diverges. These runs are labelled “div.” in the results when the value function reaches a magnitude greater than 1e9. See Table 1 of the Global Response.
>
> Thus, the fact that our algorithm avoids this issue of value function instability indicates that our algorithm prevents the value function being updated on transitions to out-of-distribution state-action pairs. This is because we do not see an alternative reason for the instability of the ablated algorithms, which operate using the same dataset and hyperparameters. Therefore, this is a strong indication that our algorithm mitigates the issue of distributional shift,
>
> [4] Kumar, Aviral, et al. "Stabilizing off-policy Q-learning via bootstrapping error reduction." NeurIPS, 2019.
>
> ### Question 4.
>
> Thank you for pointing out this typo, we will fix this.

---

> > ### Comment · Reviewer_8yT3 · 2023-08-12
> >
> > I would like to thank the authors for providing clear explanations and responses to each of my questions, this is a responsible and persuasive rebuttal.
> > I believe that the updated ablation experiment section is more complete compared to before, and the addition of baselines has made the experimental results more convincing.
> > Additionally, the authors have acknowledged the limitations of their work and have listed potential solutions for future study.

---

### Official Review · Reviewer_uPGp · 2023-07-07

**Soundness:** 3 good
**Presentation:** 3 good
**Contribution:** 3 good
**Rating:** 7
**Confidence:** 2

**Summary:**

This paper considers the problem of offline reinforcement learning for risk-averse decision-making with the distributional shift. The core insight of this paper is to incorporate epistemic uncertainty (from the distributional shift) and aleatoric uncertainty (from usual statistical errors) together and develop a way to simply penalize high uncertainty, which can handle both problems simultaneously.

**Strengths:**

I think the biggest strength of this paper is the novel idea of combining both epistemic uncertainties from the distributional shift and aleatoric uncertainty together and using a simple risk-aversion algorithm to handle both problems. I think this key idea can be applied not only to offline reinforcement learning but also to other areas where the distributional shifts are key, e.g., predictions and causal inference problems.

**Weaknesses:**

I do not think there is any obvious weakness in this paper. I only have some suggestions below.

**Questions:**

**How to choose the distribution over MDPs to capture the epistemic uncertainty?**

One of the key ideas in this paper is to use the distribution over MDPs to represent the epistemic uncertainty over the real environment. As far as I understand, on page 7 (Implementation Details), the authors suggest using an ensemble of neural networks to estimate different distributions over the MDPs. It seems to me that this choice of model classes is actually the fundamentally important part of capturing epistemic uncertainty. For example, if we include a large number of different neural networks (e.g., more than 100), the epistemic uncertainty is guaranteed to be larger than cases when we include a smaller number of neural networks (e.g., 5). **But, without an explicit model or assumption about the distributional shifts, how can researchers choose what models to estimate MDPs for approximating the epistemic uncertainty?**

**In Section 5 (Experiments), how did the authors choose the model class for capturing the epistemic uncertainty?**

I think this is an important question as there is an inherent tradeoff --- if we include more models, we can be more robust to a wide range of potential distributional shifts, but if we include too many models, we might be too conservative. **Is there any theoretical guidance about how to approximate the epistemic uncertainty?**



**Limitations:**

The paper clarifies its limitation clearly.

---

> ### Author Rebuttal · Authors · 2023-08-09
>
> Thank you for the valuable time you have spent reviewing our paper.
>
> We agree that deciding how to represent epistemic uncertainty in deep learning is an important problem, and a key aspect of our work.
>
> We used a small ensemble of 5 neural networks to ensure that our work is a fair comparison to previous works, which also use a small ensemble. MOPO [1], COMBO [2], and RAMBO [3] each use an ensemble of 5 models, and MOReL [4] uses an ensemble of 4 models. Like our work, MOPO and MOReL use these small ensembles to estimate epistemic uncertainty.
>
> We did conduct experiments using a larger ensemble of 15 models. These results can be found in Table 7 in Appendix C.4. We found that the results obtained with a larger ensemble are similar to our main results which utilise the small ensemble.
>
> The influence of the size of the model ensemble on the uncertainty estimate *depends on the type of uncertainty estimate used*. The recent work [5] analyses this empirically in the context of offline RL. The variance or standard deviation between the members of the ensemble is fairly stable as the size of the ensemble changes - see Figure 2 of [5]. However, other metrics, such as the maximum disagreement between any two members of the ensemble, increase significantly as the size of the ensemble increases (again, see Figure 2 of [5]). In the latter case, we would indeed expect the uncertainty estimate to be highly sensitive to the number of models in the ensemble.
>
> In our approach, the risk-sensitive Bellman equation (Equation 4) penalises the *variability* in samples drawn from the ensemble. As stated above, the variance between ensemble members is quite stable as the size of the ensemble increases. Thus, we do not expect our approach to be highly sensitive to the size of the ensemble, as supported by the results in Table 7.
>
> Furthermore, we present a theoretical analysis in Proposition 1 in the response to Reviewer RPWe that thoroughly analyses how our approach mitigates epistemic and aleatoric uncertainty in an ensemble of models.
>
> [1] Yu, Tianhe, et al. "Mopo: Model-based offline policy optimization." NeurIPS (2020).
>
> [2] Yu, Tianhe, et al. "Combo: Conservative offline model-based policy optimization." NeurIPS (2021).
>
> [3] Rigter, Marc, Bruno Lacerda, and Nick Hawes. "Rambo-rl: Robust adversarial model-based offline reinforcement learning." NeurIPS (2022).
>
> [4] Kidambi, Rahul, et al. "Morel: Model-based offline reinforcement learning." **NeurIPS (2020).
>
> [5] Lu, Cong, et al. "Revisiting Design Choices in Offline Model-Based Reinforcement Learning." ICLR, 2022.

---

> > ### Comment · Reviewer_uPGp · 2023-08-21
> >
> > Thank you so much for the rebuttal and your detailed clarification. These answered my questions well!

---

### Official Review · Reviewer_RPWe · 2023-07-10

**Soundness:** 3 good
**Presentation:** 3 good
**Contribution:** 3 good
**Rating:** 6
**Confidence:** 5

**Summary:**

One Risk to Rule Them All (1R2R) is an offline-RL method that seeks to reduce aleatoric and epistemic uncertainty. Their method is simple; the authors introduce the notion of risk in the bellman update by adding learned adversarial perturbation to the learned transition dynamics models in the model-based setting. This is different from prior risk-based methods which adversarially perturbate the value function as opposed to the TD function.

**Strengths:**

- The narrative/flow is clear; this work focuses on minimizing both aleatoric and epistemic uncertainty, both of which hamper offline RL agent performance.
- Proposed methodology is simple, just add a learned adversarial perturbation to TD model.
- Figure 1 is an excellent figure that illustrates that the learned risk-aware value function penalizes both uncertainties. Yet, it is difficult to absorb/fully understand, and it might be better to split it into two subfigures. For example, highlight regions with different colors outside of dataset and regions where  TD uncertainty is high. (It is also nice to have to compare the value function to CQL's too, to show that CQL might be aleatoric unaware)
- Evaluation is thorough and evaluated over difficult environments with large continuous action spaces.

**Weaknesses:**

- The intro was not clear. At the end, it says that epistemic uncertainty is avoided through model-ensemble variance. However, the analogy to aleatoric uncertainty was not made clear. It might be better to straight-up state that risk-averse RL applied in the offline setting reduces both uncertainties for reasons X and Y.
- Authors claims method is simpler than prior work due to only considering risk; however, the risk incorporates model-based approaches, which is much more complicated and much harder to work in practice.
- Why does risk-averse RL reduce aleatoric uncertainty? (good to show math in appendix) It seems to be a miraculous cure to a problem.
- Methodologically, this paper is not novel; it seems more like A + B, where A = model-based ensemble in MOPO and B = risk-averse RL objective in CvAR, applied in the offline RL setting. The real novelty seems to stem from identifying applicability of risk in offline RL.
- The Risk background is difficult to understand; suggest removing it and immediately diving into the Bellman risk-sensitive representation.
- It'd be nice to have a figure illustrating the final algorithm architecture and well as typical reward x training iteration graphs in the evaluation.
- Where is the evaluation against MOPO/COMBO, another model-based offline RL method? It's be worthwhile to show that this work is the SOTA model-based offline RL approach.
- Suggesting removing one of the evaluation environments and focusing more on ablation and analysis.
- Why does 1R2R flounder on expert dataset (medium expert)?
- Another way to avoid such uncertainty is the field of safe RL, which seeks to minimize constraint violations while maximizing environmental reward (Recovery RL). There is a lot of prior work in this field and is somehow obfuscated in this paper.
-Nit: There are several typos, including in Algorithm 1 line 9,  and Section 3 Static and Dynamic Risk MDP formulation.

**Questions:**

N/A

**Limitations:**

- Unrelated, it would be nice to have a model-free version, as I'd generally avoid model-based approaches (as does industry). For example, if you add the adversarial perturbation to the distributional value function (instead of the learned TD function), despite prior work.
- I'd upgrade this paper to strong accept (as possibly higher) if it explains WHY adding risk-aversion reduces epistemic and aleotoric uncertainty, as this is a very beautiful insight. Bonus if there is math.

---

> ### Author Rebuttal · Authors · 2023-08-09
>
> Thank you for the time you have spent reviewing our paper. We respond to your main comments below. We are unable to respond to all comments due to space constraints.
>
> ### Why does our approach reduce both epistemic and aleatoric uncertainty?
>
> In Proposition 1, and the explanation that follows, we provide mathematical justification for why our approach reduces both epistemic and aleatoric uncertainty. We will add this to the final paper.
>
> In Proposition 1, we assume that each transition function in the ensemble is Gaussian, with standard deviation $\sigma_A$ representing the aleatoric uncertainty. We also assume that the distribution of the means of each member of the ensemble is Gaussian with standard deviation $\sigma_E$ (representing epistemic uncertainty). Proposition 1 and the corollary that follows demonstrate that our approach penalises taking actions that have *either* large $\sigma_A$ or $\sigma_E$. We discuss Proposition 1 in detail below.
>
> ---
>
> **Proposition 1:**  Consider some state $s$ in a 1D state space, and some action $a$. Assume 1) that there is an ensemble of $M$ Gaussian transition functions, each denoted $T_i$ with mean $\mu_i$ and standard deviation $\sigma_A$: $\\{T_i(s' | s, a)  = \mathcal{N}(\mu_i, \sigma_A^2) \\}_{i=1}^M$; and 2) that the mean of each Gaussian, $\mu_i$, is normally distributed with mean $\mu_0$ and standard deviation $\sigma_E$: $\mu_i \sim \mathcal{N}(\mu_0, \sigma_E^2)$. According to Eq. 5, to sample a successor state $s'$ from $\overline{T}$ we must sample a transition function $T_i$ from the ensemble, and then sample $s' \sim T_i(s' | s, a)$.
>
> Assume that the value of the successor state is linear around $\mu_0$ with some linearity constant, $K$:
>
> $$
> V(s') = V(\mu_0) + K (s' - \mu_0)
> $$
>
> Then $V(s')$ is distributed according to: $V(s') \sim \mathcal{N}\Big(\mu = V(\mu_0), \sigma^2 = K^2(\sigma_A^2 + \sigma_E^2) \Big)$.
>
> **Proof**: The joint probability of sampling an ensemble member with mean, $\mu_i$, and then sampling a successor state, $s'$, is:
>
> $$
> P(\mu_i, s') = P(\mu_i) P(s'\ |\ \mu_i).
> $$
>
> We marginalise over  $\mu_i$:
>
> $$
> P(s') = \int_{-\infty}^\infty P(\mu_i) P(s'\ |\ \mu_i) \mathrm{d} \mu_i
> $$
>
> $P(\mu_i)$ and $P(s'\ |\ \mu_i)$ are both Gaussian. Substituting the appropriate probability density functions we have that
>
> $$
> P(s') = \int_{-\infty}^\infty \frac{1}{\sigma_E \sqrt{2\pi} } \exp^{- \frac{(\mu_i - \mu_0 )^2}{2 \sigma_E^2}}  \frac{1}{\sigma_A \sqrt{2\pi} } \exp^{- \frac{(s' - \mu_i )^2}{2 \sigma_A^2}} \mathrm{d} \mu_i \\
> = \frac{1}{\sqrt{\sigma_A^2 + \sigma_E^2} \sqrt{2\pi} } \exp^{-\frac{(s' - \mu_0 )^2}{2 \sqrt{\sigma_E^2 + \sigma_A^2}}} = \mathcal{N}(\mu_0, \sigma_E^2 + \sigma_A^2).
> $$
>
> Thus, $s'$ is normally distributed with mean $\mu_0$ and standard deviation $\sqrt{\sigma_E^2 + \sigma_A^2}$. The random variable $K(s' - \mu_0)$ is therefore normally distributed with mean 0 and standard deviation $|K|\sqrt{\sigma_E^2 + \sigma_A^2}$. Finally, $V(s') = V(\mu_0) + K (s' - \mu_0)$ is also normally distributed with mean $V(\mu_0)$ and standard deviation $|K|\sqrt{\sigma_E^2 + \sigma_A^2}$. $\square$
>
> ---
>
> In Proposition 1, $\sigma_E$ defines the level of disagreement between the members of the ensemble, and therefore represents the level of epistemic uncertainty. $\sigma_A$ represents the aleatoric uncertainty. The proposition tells us that if either $\sigma_A$ or $\sigma_E$ are high, then there is high variability over the value of the successor state when sampling from the ensemble (i.e. sampling from $\overline{T}$ in Eq. 5).
>
> Risk measures penalise high variability. The risk-sensitive Bellman equation (Eq. 4) applies a risk-measure the value over the successor state. Therefore, applying Eq. 4 to samples from the ensemble penalises executing state-action pairs for which either $\sigma_A$ or $\sigma_E$ is high. Thus, our approach penalises actions with *either* high aleatoric or epistemic uncertainty. It therefore favours choosing actions that have *both* low aleatoric and epistemic uncertainty.
>
> The following corollary uses the example of conditional value at risk (CVaR) to show how the value computed in the risk-sensitive Bellman equation decreases as either $\sigma_A$ or $\sigma_E$ increase.
>
> ---
>
> **Corollary:** Under the assumptions in Proposition 1, the CVaR at confidence level $\alpha$ of the value of the successor state is:
>
> $$
> \mathrm{CVaR}_\alpha\big(V(s')\big) = V(\mu_0) - \frac{|K|\sqrt{\sigma_E^2 + \sigma_A^2}}{\alpha \sqrt{2 \pi}}  \exp ^ {-\frac{1}{2}( \Phi^{-1}(\alpha) )^2}
> $$
>
> where $\Phi^{-1}$ is the inverse of the standard normal CDF.
>
> **Proof:** This follows directly from Proposition 1 and the formula for the CVaR for a Gaussian random variable [1].
>
> [1] Khokhlov, V.. "Conditional value-at-risk for elliptical distributions." European Journal of Economics and Management (2016).
>
> ---
>
> ### Extra Figures
>
> In Fig 1 of the Global Response, we have added a summary figure. We will add this to the paper. We have also included example plots of the performance vs training iterations. We will add these plots for all datasets to the final paper.
>
> ### Extra Ablation
>
> We have added an **additional ablation**. See the response to Reviewer 8yT3 for details.
>
> ### MOPO/COMBO
>
> We have **expanded the results** to include MOPO, COMBO, and ATAC. See the response to Reviewer unPm for details.
>
> ### Medium-Expert Datasets
>
> Poor performance of model-based approaches on high-quality datasets is an established issue [1, 2]. The cause of this is an open question in the community which we wish to investigate in future work. Model-based methods are generally strongest on noisy sub-optimal data [1].
>
> [1] Lu, C, et al. "Challenges and opportunities in offline reinforcement learning from visual observations." TMLR, 2023
>
> [2] Rigter, M, et al. "Robust adversarial model-based offline reinforcement learning." NeurIPS, 2022
>
> ### Related Work
>
> We will expand the related work to discuss constrained RL and recovery RL.

---

### Author Rebuttal · Authors · 2023-08-09

Thank you for your reviews.

Please find attached the Global Response. In the attachment, we provide updated results which include an **additional ablation**, as well as comparisons to **additional baseline algorithms**: MOPO, COMBO, and ATAC.

We also include additional figures. The summary figure (Figure 1 in the global response) will be added to the final version of the paper. We have also provided examples of performance vs training iterations (Figure 2 in the global response). These figures will be added for all datasets in the final version.

Finally, we have also added stronger **theoretical justification** for why our approach reduces both aleatoric and epistemic uncertainty. Please see Proposition 1 in the response to Reviewer RPWe. This theoretical motivation will also be added to the final version of our paper.

Please let us know if there is anything further that you would like to discuss!

Kind regards,

The authors

---

### Decision · Program_Chairs · 2023-09-21

**Decision:**

Accept (poster)

**Comment:**

All the reviewers like the work and see it above the bar. There are questions/concerns about parts of the paper that some were answered during the rebuttal phase. It would be great if the authors revise their work to properly address the reviewers' comments when they prepare the camera-ready version.